# Fairness under Competition

**Ronen Gradwohl**
Department of Economics
University of Haifa
rgradwohl@econ.haifa.ac.il

**Eilam Shapira**
Faculty of Data and Decision Sciences
Technion – Israel Institute of Technology
eilam.shapira@gmail.com

**Moshe Tenneholtz**
Faculty of Data and Decision Sciences
Technion – Israel Institute of Technology
moshet@ie.technion.ac.il

## Abstract

Algorithmic fairness has emerged as a central issue in ML, and it has become standard practice to adjust ML algorithms so that they will satisfy fairness requirements such as Equal Opportunity. In this paper we consider the effects of adopting such fair classifiers on the overall level of *ecosystem fairness*. Specifically, we introduce the study of fairness with competing firms, and demonstrate the failure of fair classifiers in yielding fair ecosystems. Our results quantify the loss of fairness in systems, under a variety of conditions, based on classifiers' correlation and the level of their data overlap. We show that even if competing classifiers are individually fair, the ecosystem's outcome may be unfair; and that adjusting biased algorithms to improve their individual fairness may lead to an overall decline in ecosystem fairness. In addition to these theoretical results, we also provide supporting experimental evidence. Together, our model and results provide a novel and essential call for action.

## 1 Introduction

Algorithmic decision-making systems are not immune to human prejudices. This has been demonstrated by ample empirical evidence: For example, the use of algorithmic decision-making in determining loan approval and interest rates has led to minority applicants facing higher loan rejection rates and higher interest rates than non-minority applicants (Bartlett et al., 2022; Fuster et al., 2022), and credit card issuers offering lower credit limits for women than men (Gupta, 2019; Telford, 2019). Additional contexts in which such algorithmic bias has been documented are as diverse as job recruitment (Doleac and Hansen, 2016), insurance premiums and payouts (Angwin et al., 2017), college admissions (Baker and Hawn, 2022; Gándara et al., 2024), and more (O'Neil, 2016).

To contextualize, consider a bank or other financial institution that issues loans and is subject to regulatory oversight. The bank employs historical data to train a classifier—a decision rule—to determine loan eligibility, with the aim of maximizing profit through successful loan repayments. However, such classifiers frequently exhibit biases against protected groups, such as ethnic minorities. In response, regulators can mandate fairness restrictions to mitigate these disparities. Bias is not only normatively problematic, but also has tangible adverse effects on the utility of affected individuals and, more broadly, on the welfare of the disadvantaged group.

The regulator's goal in imposing fairness adjustments to classifiers is thus to enhance the welfare of historically disadvantaged populations. One principled approach to fairness adjustment involves specifying a utility function, and requiring the classifier to ensure equal welfare across groups.

39th Conference on Neural Information Processing Systems (NeurIPS 2025).

Prominent fairness criteria such as Demographic Parity (DP) (Agarwal et al., 2018; Dwork et al., 2012), as well as Equal Opportunity (EO) and Equalized Odds (ED) (Hardt et al., 2016) can be interpreted as special cases within this welfare-based framework. For instance, DP corresponds to a utility specification where each approved applicant gains a utility of one and zero otherwise. EO and ED, correcting for some criticisms about DP, does the same but only for the restricted set of deserving applicants.

Fairness interventions are intended to promote equity. While even widely adopted constraints like DP and EO may lead to suboptimal long-term outcomes (Liu et al., 2018), common wisdom holds that proper fairness adjustments of ML algorithms is an essential requirement.

But is the adoption of fair algorithms by firms sufficient to guarantee a *fair AI ecosystem*, in which multiple firms interact? Recent work in AI has recognized that AI systems should be evaluated in the context of multi-agent system, in which several stakeholders are active (Kurland and Tennenholtz, 2022; Boutilier et al., 2024). In lending, multiple financial institutions underwrite loans and issue credit cards; in employment, many firms vie for the same pool of job applicants; in insurance, numerous insurance companies offer policies; and in education, a large number of colleges compete over the same cohorts of students. In such multi-firm environments, equity is not about whether a single firm's decisions are fair, but whether there is *overall discrimination* in the number and quality of offers issued and opportunities afforded to loan, job, insurance, and college applicants.

In the world of algorithmic fairness, this translates into a novel question. Suppose there are several competing firms, each of which adopts a fair classifier. Does the adoption of fair classifiers induce a fair ecosystem, in which there is no overall discrimination?

In this paper, we introduce the study of *fairness with competing firms*. We provide fundamental definitions, and demonstrate the failure of fair classifiers in obtaining fair ecosystems. In particular, we show that even if competing classifiers are individually fair, the ecosystem's outcome may be unfair; and that adjusting biased algorithms to improve their individual fairness may lead to an overall *decline* in ecosystem fairness.

**Contributions** We begin in Section 2 by developing a model of ecosystem fairness. We suppose there are multiple lenders who make loan offers to borrowers; however, the model applies analogously to job candidates and employers, insurance buyers and insurers, or student applicants and colleges. For this model we define our main notion of fairness under competition, a competition analogue of EO that we call Equal Opportunity under Competition (EOC). The level of EOC here is a measure of how far the ecosystem is from satisfying EOC, where a lower level implies higher ecosystem fairness. We also define a second, welfare-based version of EOC. An ecosystem satisfies this second notion if the welfare of different groups of borrowers is equalized when the lenders use their classifiers to make loan offers.

In Section 3 we turn to study EOC. Our main results are that, even when the individual classifiers satisfy EO, the ecosystem may be far from EOC. We quantify how far, based on model primitives, and provide worst-case bounds. In particular, we identify two distinct forces under which EO classifiers do not satisfy EOC. The first force, analyzed in Section 3.1, arises from differences in the correlations between the classifiers on the protected groups. The second force, analyzed in Section 3.2, arises when the lenders' pools of potential borrowers are overlapping but not identical. We quantify the extent to which each force decreases ecosystem fairness.

To illustrate the first force, consider the following example:

**Example 1** There are two lenders and two groups of borrowers. Each lender has vast, distinct data on group 1, and trains a classifier to predict loan repayment. Neither lender has sufficient data on group 2, so both outsource to a third-party, who provides each lender with an (identical) classifier. Each lender uses her own and the third-party classifiers to offer loans to individuals predicted to repay the loan.

In this example, if the lenders' and third-party's classifiers have the same accuracies, then each lender's decisions satisfy EO. Note, however, that the lenders' predictions on group 2 are identical, as they are using the same classifier, and so the Pearson correlation between them is high. On the other hand, since the lenders use different classifiers with different training data on group 1, their respective predictions on this group have lower correlation. In Proposition 1 and Corollary 1 we show that this difference in correlations leads to a positive level of EOC, of the same order-of-magnitude as

the classifiers' false-negative rates. Hence, competition between EO classifiers leads to a non-EOC outcome.

To illustrate the second force, consider the following example:

**Example 2** There are two lenders and two groups of borrowers. Each lender serves all applicants of group 1 but only a subset of applicants of group 2, perhaps because the lenders have lower market penetration in the latter population. For served applicants, lenders use an EO classifier to predict loan repayment and offer loans to individuals predicted to repay.

In this example, the overlap between applicants served by the two classifiers in group 1 is higher than in group 2. In Proposition 4 and Corollary 3 we show that such a difference in overlaps leads to a positive level of EOC, again of the same order-of-magnitude as the classifiers' false-negative rates, and again despite the fact that individual classifiers *do* satisfy EO.

Section 3 focuses on EOC when there are two lenders. However, in Sections 3.1.1 and 3.1.2 we extend some results to the welfare-based notion of fairness and to an arbitrary number of lenders.

In Section 4 we compare competition between classifiers that are not EO with competition between the same classifiers after undergoing a fairness adjustment (making them EO). We describe two examples in which the ecosystem is EOC *before* the fairness adjustment, but *not* EOC afterwards. These examples show that imposing fairness adjustment on individual classifiers can lead to a decline in ecosystem fairness.

Next, in order to explore the empirical prevalence of our theoretical results, in Section 5 we describe several experiments we ran on Lending Club loan data. In these experiments we compared the level of EOC before and after fairness adjustments under several variations. Our results indicate that, surprisingly, imposing fairness adjustments *often* leads to a higher level of EOC, and so lower ecosystem fairness.

Finally, in Appendix B we extend our results to other notions of fairness, namely, ED and DP.

**Related literature** Our paper is part of a large and growing literature on fairness in machine learning (see, e.g., Chouldechova and Roth, 2018; Mehrabi et al., 2021; Pessach and Shmueli, 2022; Barocas et al., 2023, among others). The literature includes a plethora of demonstrations of bias in machine learning, a large number of fairness metrics, and many algorithms for bias mitigation (the last of which is surveyed extensively by Hort et al., 2024).

Within this large literature, our paper fits within a subset of papers that question whether fair machine learning algorithms actually achieve fairness (see Ruggieri et al., 2023, for a survey). One paper related to ours in this vein is that of Liu et al. (2018), who show that imposing fairness criteria may have adverse long-term effects. This can be seen as illustrating the failure of imposed fairness criteria due to temporal or sequential effects. Our paper, in contrast, can be seen as illustrating the failure due to competition or parallel effects.

Our research is more closely related to papers that examine fairness in settings with multiple classifiers. Two such papers, Bower et al. (2017) and Dwork and Ilvento (2019), also discuss the insight that fairness of individual classifiers does not imply fairness of a system composed of multiple classifiers. Our paper differs along multiple dimensions. First, in terms of motivation, these papers ask when a central platform (e.g., an ad platform) will be fair when handling a task that consists of several subtasks (advertisers on the platform), where each subtask is required to be fair. Our motivation is more about competing firms in a decentralized market, where the joint activity determines the user's utility. More specifically, Dwork and Ilvento (2019) largely focus on individual fairness. They do have some extensions to group fairness, and their main results here are to show that there exist distributions for which individual fairness does not imply joint fairness. However, in their model the classifiers are assumed to always be independent, and they cannot capture the correlations between classifiers that lead to ecosystem unfairness. In addition, their notion of group fairness does not include EO. Bower et al. (2017) do focus on EO, but here the main difference is that classifiers are composed sequentially: one classifier makes a prediction or decision, the outcome is then passed on to the next classifier, and so on. In our setting, in contrast, the classifiers run in parallel. Finally, although both papers contain the insight that individual fairness does not suffice for joint fairness, in our paper we also identify the forces that lead to the joint unfairness, and quantify the extent of this unfairness as a function of correlations and overlaps.

Another somewhat related paper is that of Ustun et al. (2019), who compare the benefits to protected groups when there is one classifier with imposed fairness criteria, as opposed to many group-tailored individual classifiers. Although the latter case involves multiple classifiers, individuals are not classified by more than one, and so there is no competition between the classifiers.

Our paper is also part of a smaller literature on competition between machine learning algorithms (see, e.g., Ben-Porat and Tennenholtz, 2017; Feng et al., 2022; Jagadeesan et al., 2023). While that literature demonstrates that competition has some counter-intuitive effects, it has not examined issues surrounding fairness.

Finally, Example 1 above is related to the insight of Kleinberg and Raghavan (2021), who show that algorithmic monoculture—the use of the same algorithms by different firms—can lead to a decrease in utilities. In Example 1 the use of the third-party classifier by the two firms leads to a difference in correlations, which can then lead to ecosystem bias.

## 2 Model

There are two types of players, borrowers and lenders. Each borrower is characterized by a triplet $(x, a, y) \in X \times A \times Y$, where $x$ is a vector of observable features, $a$ is an observable protected attribute (group membership), and $y$ is an unobservable outcome variable. For simplicity, assume $A = Y = \{0, 1\}$. There is an underlying distribution $\mathcal{D}$ over $X \times A \times Y$. We slightly abuse notation by using $X$, $A$, and $Y$ as the sets of features, protected attributes, and outcome variables, and also as random variables with joint distribution $\mathcal{D}$ over the respective sets. Throughout, we think of each $(x, a, y) \in X \times A \times Y$ as an individual borrower.

There is a set $L$ of lenders. Each lender $\ell \in L$ decides whether or not to offer a loan to each borrower based on observables $(x, a)$. To do so the lender uses a potentially randomized classifier $c_\ell : X \times A \mapsto \{0, 1\}$, where $c_\ell(x, a) = 1$ if the lender offers borrower $(x, a)$ a loan, and $c_\ell(x, a) = 0$ otherwise. The unobservable outcome variable $Y$ signifies whether or not the borrower repays the loan, and we call borrowers with $y = 1$ *deserving* borrowers. Naturally, lenders prefer to offer loans only to deserving borrowers. Hence, perfect classifiers are ones where $c_\ell(x, a) = y$ for each borrower $(x, a, y)$. In general, however, firms' classifiers are not perfect. For a given classifier $c_\ell$, denote its false-negative rate by $\beta_\ell = \Pr[c_\ell(X, A) = 0 | Y = 1]$ and its false-positive rate by $\alpha_\ell = \Pr[c_\ell(X, A) = 1 | Y = 0]$. Conversely, $1 - \beta_\ell$ and $1 - \alpha_\ell$ are the classifier's true-negative and true-positive rates, respectively.

Borrowers' utilities are described by a function $v : X \times A \times Y \times \mathbb{N} \mapsto \mathbb{R}$, where $\mathbb{N}$ is the set of non-negative integers. $v(x, a, y, r)$ is the utility of a borrower $(x, a, y)$ when she receives $r$ offers.[1]

**Fairness with a single lender**   We begin by stating two notions of fairness for the setting of a single classifier.[2] The first is the popular notion of EO (Hardt et al., 2016), which requires the classifier's false-negative rates to be equal across groups. The second is based on the welfare-equalizing notion of Ben-Porat et al. (2021) that we call $v$-EO, and which requires the expected utilities of deserving borrowers to be equal across groups.

EO:   The EO level of $c_\ell$ is $|\mathrm{E}[c_\ell(X, A) | Y = 1, A = 0] - \mathrm{E}[c_\ell(X, A) | Y = 1, A = 1]|$, the difference in the fraction of offers made to each of the groups' deserving borrowers. A classifier is EO if its EO level is 0.

$v$-EO:   For a given utility function $v$ of the borrowers and classifier $c_\ell$ of a lender, denote by $W_{v,c_\ell}(a) = \mathrm{E}[v(X, A, Y, c_\ell(X, A)) | Y = 1, A = a]$ the welfare of deserving borrowers from group $a$. The $v$-EO level of a classifier $c_\ell$ is $|W_{v,c_\ell}(0) - W_{v,c_\ell}(1)|$, the difference in welfare between the deserving borrowers of the two groups under classifier $c_\ell$.

**Fairness with multiple lenders**   We now generalize the definitions of fairness to a setting with multiple competing lenders, EO under competition (EOC) and $v$-EO under competition ($v$-EOC). Fix a utility function $v$ and classifiers $c = (c_1, \ldots, c_{|L|})$, and denote by $R(x, a) = \sum_{\ell \in L} c_\ell(x, a)$ the num-

---

[1] More generally, borrowers' utilities could also depend on *which* lenders extended offers. Our definitions and results apply to this more general setting as well.

[2] These notions are the foci of the paper, but in Appendix B we extend our results to other notions.

ber offers made to a borrower with observables $(x, a)$. Also, denote by $d(x, a) = \Pr[R(x, a) \geq 1]$, the probability that at least one lender offers a borrower with observables $(x, a)$ a loan.

EOC: The EOC level of classifiers $c$ is $|\mathrm{E}[d(X, A)|Y = 1, A = 0] - \mathrm{E}[d(X, A)|Y = 1, A = 1]|$, the difference between the two groups in the fraction of deserving borrowers who obtained at least one offer. Classifiers are EOC if their EOC level is 0.

$v$-EOC: For a given utility function $v$ of the borrowers and classifiers $c$ of the lenders, denote by $W_{v,c}(a) = E[v(X, A, Y, R(X, A))|Y = 1, A = a]$. The $v$-EOC level of classifiers $c$ is then $|W_{v,c}(0) - W_{v,c}(1)|$.

In some of our analyses we will impose several assumptions. First, we sometimes focus on the case of two lenders. Second, we sometimes focus on the case in which borrowers do not care how many offers they obtain, as long as they obtain at least one. In this case, their preferences satisfy the following:

**Assumption 1 (0-1 preferences)** $v(x, a, y, r) = 1$ *if* $r \geq 1$ *and* $v(x, a, y, 0) = 0$, $\forall x, a, y$.

Observe that under Assumption 1, EO is equivalent to $v$-EO and EOC is equivalent to $v$-EOC. For more general utility functions EO and EOC may differ from $v$-EO and $v$-EOC.

## 3 Equal Opportunity under Competition

In this section we ask the following question: Given two EO classifiers, are they guaranteed to be EOC? We show that the answer is negative, and identify two distinct forces that drive this result. We also quantify the level of EOC, and bound the worst case. The first force, analyzed in Section 3.1, arises from differences in the correlation between the classifiers on the two protected groups. The second force, analyzed in Section 3.2, arises when each classifier serves only a subset of the borrowers.

### 3.1 Correlations between classifiers

Fix two EO classifiers $c_1$ and $c_2$ with false-negative rates $\beta_1$ and $\beta_2$. We define two Bernoulli random variables that capture the true positives of the classifiers on the two groups $a \in A$: For each $a \in A$ and each $\ell \in \{1, 2\}$, let the Bernoulli random variable

$$B_\ell^a \equiv c_\ell(X, A)|_{(Y=1, A=a)}.$$

Each $B_\ell^a$ is the output of classifier $c_\ell$ on random instances with $A = a$ and $Y = 1$, and so $\mathrm{E}[B_\ell^a] = 1 - \beta_\ell$ is the true-positive rate of classifier $c_\ell$. Denote by $\sigma_\ell = \sqrt{\beta_\ell(1 - \beta_\ell)}$ the standard deviation of $B_\ell^0$, and note that, because $c_\ell$ is EO, $\sigma_\ell$ is also the standard deviation of $B_\ell^1$. Finally, for each $a$ let $\rho^a$ denote the Pearson correlation coefficient between $B_1^a$ and $B_2^a$. The Pearson correlation captures the extent to which the true positives of the classifiers correlate with one another.

We now quantify the EOC level of two classifiers using these correlation coefficients, and determine the worst case.

**Proposition 1** *For two EO classifiers $c_1$ and $c_2$ with false-negative rates $\beta_1$ and $\beta_2$, the level of EOC is $\sigma_1 \cdot \sigma_2 \cdot |\rho^0 - \rho^1|$. In the worst case, the level of EOC is $\min\{\beta_1, \beta_2\} - \max\{0, \beta_1 + \beta_2 - 1\}$.*

The intuition is the following. Under high (resp., low) correlation, if a borrower is misclassified by one classifier, she is likely also (resp., not) misclassified by the other. In the low correlation case, each deserving borrower intuitively has "two chances" to get an offer, whereas in the high correlation case she only has "one chance". If in one group deserving borrowers get two chances and in the other they only get one, then in the former there is a lower chance of being misclassified.

The proofs of this and all other propositions are deferred to Appendix A due to space constraints. Proposition 1 yields the following simple corollary.

**Corollary 1** *If $\beta = \beta_1 = \beta_2$ and $\beta \leq 1/2$, then the worst-case level of EOC is $\beta$.*

An implication of Proposition 1 and Corollary 1 is that the more accurate the classifiers get, the lower the worst-case level of EOC. Thus, one way to minimize the EOC is to train more accurate classifiers, and so this desideratum is aligned with the general goal of machine learning.

We can also use Proposition 1 to analyze Example 1.

**Example 1 continued**   Suppose all classifiers in the example—the third-party classifier, as well as the two firms' individual classifiers—have the same false-negative rate $\beta$. Thus, a firm that uses its own classifier on $A = 1$ and the third-party classifier on $A = 0$ is in practice using an EO classifier. Suppose also that the firms' individual classifiers $c_1$ and $c_2$ are uncorrelated on $A = 1$, namely, that $\rho^1 = 0$. Finally, note that, since both firms use the same classifier on $A = 0$ we also have $\rho^0 = 1$. Thus, even though each firm uses an EO classifier, the level of EOC is $\sigma_1 \sigma_2 |0 - 1| = \beta(1 - \beta)$.

### 3.1.1   Generalization: $v$-EOC

Recall that, under Assumption 1, EOC is equivalent to $v$-EOC, in which case all our results apply to the latter notion as well. In this section we study $v$-EOC when utility functions do not satisfy Assumption 1. In particular, we drop the assumption that $v(x, a, y, 1) = v(x, a, y, 2) = 1$, and instead assume the following:

**Assumption 2 (0-1-k preferences)**  $v(x, a, y, r) = k \geq 1$ *for* $r > 1$, $v(x, a, y, 1) = 1$, *and* $v(x, a, y, 0) = 0$, $\forall x, a, y$.

Assumption 1 imposes $k = 1$, but now we consider all values of $k \geq 1$. Observe that under Assumption 2, a classifier is still EO if and only if it is $v$-EO. However, for a pair of classifiers $c_1$ and $c_2$, the definitions of EOC and $v$-EOC are no longer equivalent.

We first generalize Proposition 1 to this setting, and then apply it to Example 1.

**Proposition 2**  *Under Assumption 2, for two EO classifiers $c_1$ and $c_2$ with false-negative rates $\beta_1$ and $\beta_2$, the level of $v$-EOC is $\sigma_1 \sigma_2 \left|(k - 2)(\rho^0 - \rho^1)\right|$. In the worst case, the level of $v$-EOC is $|k - 2| \cdot (\min\{\beta_1, \beta_2\} - \max\{0, \beta_1 + \beta_2 - 1\})$.*

There are a few interesting observations to note. First, in all cases except $k = 2$, the level of $v$-EOC is positive, even though both classifiers are EO (and so, under Assumption 2, also $v$-EO). Hence, our insight is, in a sense, robust. Interestingly, however, there is a qualitative difference between the case $k \in [1, 2)$ and $k > 2$, in that the group with lower utility under competition is different in the two parameter intervals. This last point is further explained at the end of this sub-section, in the context of the example.

Proposition 2 yields the following simple corollary.

**Corollary 2**  *If $\beta = \beta_1 = \beta_2$ and $\beta \leq 1/2$, then the worst-case level of $v$-EOC is $\beta \cdot |k - 2|$.*

We can also use Proposition 2 to further analyze Example 1.

**Example 1 continued**   Suppose again that all classifiers in the example—the third-party classifier, as well as the two firms' individual classifiers—have the same false-negative rate $\beta$. Thus, a firm that uses its own classifier on $A = 1$ and the third-party classifier on $A = 0$ is in practice using an EO classifier. Suppose also that the firms' individual classifiers $c_1$ and $c_2$ are uncorrelated on $A = 1$, namely, that $\rho^1 = 0$. Finally, note that, since both firms use the same classifier on $A = 0$ we also have $\rho^0 = 1$. Thus, even though each firm uses an EO classifier, the level of $v$-EOC is $\beta(1 - \beta) \cdot k - 2$. Furthermore, the utility of borrowers in group $A = 0$ is $k(1 - \beta)$, since all borrowers who receive an offer actually receive two offers. In group $A = 1$, on the other hand, the expected utility of borrowers is $k(1 - \beta)^2 + 2\beta(1 - \beta)$.

Now,
$$k(1 - \beta)^2 + 2\beta(1 - \beta) - k(1 - \beta) = \beta(1 - \beta)(2 - k).$$
This implies that the level of $v$-EOC is $\beta(1 - \beta) |2 - k|$. However, note that, if $k \in [1, 2)$ then the difference is positive, whereas if $k > 2$ then the difference is negative. This implies that, in the former case, the expected utility is higher in group $A = 1$, whereas in the latter case, the expected utility is higher in group $A = 0$.

The main insights from this section are, first, that two $v$-EO classifiers can lead to a positive $v$-EOC even for more general utility functions, and second, that whether or not the "disadvantaged" group is the one that suffers under a positive $v$-EOC depends on the utility function (i.e., whether $k \in [1, 2)$ or $k > 2$).

### 3.1.2 Generalization: more than two classifiers

Suppose that instead of two classifiers, the set $L$ contains $n$ classifiers. We provide two results. First, we characterize the worst-case EOC with $n$ classifiers. Second, we generalize the analysis of Example 1 and show that, when classifiers are uncorrelated on one group but fully correlated on the other group, the level of EOC strictly increases (and so worsens) with $n$.

**Proposition 3** *For $n$ EO classifiers $c_1, \ldots, c_n$ with false-negative rates $\beta_1, \ldots, \beta_n$, the worst-case level of EOC is $\min_{i \in L} \beta_i - \max\left\{0, \sum_{j \in L} \beta_j - 1\right\}$. If $\beta_i = \beta \leq 1 - 1/n$ for all $i$, then the worst-case level of EOC is $\beta$.*

We now analyze a further extension of Example 1, in which we show that the level of EOC is strictly increasing in the number of lenders.

**Example 1 continued** Suppose there are $n$ lenders. As in the original example, each lender has vast, distinct data on group $A = 1$, and trains a classifier to predict loan repayment. No lender has sufficient data on group $A = 0$, so both outsource to a third-party, who provides each lender with an (identical) classifier. Suppose all classifiers have the same false-negative rate $\beta$. Thus, a firm that uses its own classifier on $A = 1$ and the third-party classifier on $A = 0$ is in practice using an EO classifier. Suppose also that, on $A = 1$, the firms' individual classifiers misclassify positive instances independently of other classifiers' predictions:

$$\Pr\left[c_1(X, A) = \ldots = c_n(X, A) = 0 | A = 1, Y = 1\right] = \beta^n.$$

Additionally, since all lenders use the same classifier on $A = 0$, the probability that a positive instance from $A = 0$ is misclassified, $\Pr\left[c_1(X, A) = \ldots = c_n(X, A) = 0 | A = 0, Y = 1\right]$, is equal to $\beta$. Thus, even though each firm uses an EO classifier, the level of EOC is $\beta - \beta^n$. Observe that this is increasing in $n$.

The main insight from this section is that rather than improving the situation, increasing the number of (EO) classifiers can actually lead to a larger EOC.

## 3.2 Different sets of borrowers

Recall that each borrower is a triplet $(x, a, y) \in X \times A \times Y$, and that there is an underlying distribution $\mathcal{D}$ over $X \times A \times Y$. In this section we modify the model and suppose that each classifier $\ell$ serves only a subset $S_\ell \subseteq \mathrm{supp}(\mathcal{D})$ of borrowers. Let us assume that $S_1 \cup S_2 = \mathrm{supp}(\mathcal{D})$. In Section 3.1 we assumed that $S_1 = S_2 = \mathrm{supp}(\mathcal{D})$, but in this section we assume that $S_1$ and $S_2$ are only partially overlapping, and so that $S_1 \neq S_2$. In the extreme case, the two subsets of borrowers are disjoint.

Formally, assume each classifier faces an underlying distribution $\mathcal{D}_\ell$ of borrowers, where $\mathcal{D}_\ell$ is equal to the distribution $\mathcal{D}$ conditional on $S_\ell$. Denote by $\gamma_\ell^a = \Pr_{\mathcal{D}_\ell}[A = a]$ the fraction of borrowers belonging to group $A = a$ that are served by classifier $\ell$, and by $\gamma^a$ the fraction of borrowers of group $a$ served by both lenders. Note that $\gamma_1^a + \gamma_2^a - \gamma^a = 1$.

In general, classifiers may serve different sets of borrowers and at the same time differ in other ways—they may have different error rates on borrowers served by both lenders, and may also have correlation between them (as in Section 3.1). In order to focus on the former, however, in this section we make two simplifying assumptions. First, we assume that each classifier's false-negative rate is the same for borrowers that are served exclusively by that classifier and for those that are served by both classifiers. Second, we assume that on instances $(x, a, y) \in S_1 \cap S_2$ the classifiers are uncorrelated. Formally,

**Definition 1** *Classifiers $c_1, c_2$ with false-positive rates $\beta_1, \beta_2$ are uncorrelated if (i) for each $\ell$,*

$$\Pr\left[c_\ell(X, A) = 0 | Y = 1, (X, A, Y) \in S_1 \cap S_2\right]$$
$$= \Pr\left[c_\ell(X, A) = 0 | Y = 1, (X, A, Y) \in S_\ell \setminus S_{3-\ell}\right],$$

*and (ii) for every $(x, 1, a) \in S_1 \cap S_2$ it holds that $\Pr\left[c_1(x, a) = c_2(x, a) = 0\right] = \beta_1 \cdot \beta_2$.*

In particular, if $S_1 = S_2 = \mathrm{supp}(\mathcal{D})$ as in Section 3.1 then uncorrelated classifiers satisfy $\rho^0 = \rho^1 = 0$, neutralizing the force identified by Proposition 1. This assumption thus isolates the effect of having different sets of borrowers. We now quantify this effect, and determine the worst case.

**Proposition 4** *For two uncorrelated EO classifiers $c_1$ and $c_2$ with false-negative rates $\beta_1$ and $\beta_2$, the level of EOC is*

$$\left| (\gamma_2^0 - \gamma_2^1)\beta_1 + (\gamma_1^0 - \gamma_1^1)\beta_2 + (\gamma^1 - \gamma^0)\beta_1\beta_2 \right|.$$

*In the worst case, the level of EOC is* $\max\{\beta_1, \beta_2\} - \beta_1\beta_2$.

The intuition for Proposition 4 is similar to that of Proposition 1, with high (resp., low) overlap in the former playing the same role as high (resp., low) correlation in the latter. To more clearly see the effect of different sets of served borrowers, consider the following corollary:

**Corollary 3** *For two uncorrelated EO classifiers $c_1$ and $c_2$ with false-negative rates $\beta_1 = \beta_2 = \beta$, the level of EOC is $\beta \cdot (1 - \beta) \cdot \left| \gamma^0 - \gamma^1 \right|$.*

Thus, a large EOC occurs when there is a large imbalance in the respective sizes of the overlaps in served borrowers between the two groups, $\gamma^0$ and $\gamma^1$.

## 4   Harmful fairness adjustment

In trading off fairness and welfare (accuracy), it is quite intuitive that increasing the fairness properties of a classifier decreases its welfare properties. In this section, however, we show that increasing the fairness properties of individual classifiers may lead to decreases in the *fairness* properties under competition. In particular, we show that the level of EOC can *worsen* following fairness-*improving* post-processing of the individual classifiers.

In the following, we provide two theoretical examples where this occurs. The first is based on an imbalance in correlations, as in Section 3.1, and the second on an imbalance in the sets of served borrowers, as in Section 3.2. While the examples are stylized, in Section 5 we show that not only is this phenomenon empirically plausible, it is even likely.

For the following examples, we assume that the fairness adjustment is derived via post-processing (Hardt et al., 2016): Given a learned classifier $c$, the EO classifier $\tilde{c}$ is *derived* from $c$, namely, it depends only on $A$ and the predictions $c(X, A)$. We also assume that $\tilde{c}$ minimizes squared-loss relative to all such derived EO classifiers. These simplifying assumptions imply the following lemma:

**Lemma 1** *Fix a classifier with false-negative rates $\beta^0$ on group $A = 0$ and $\beta^1$ on group $A = 1$. Then for each $\beta \in \{\beta^0, \beta^1\}$ there exists a distribution $\mathcal{D}$ and false positive-rates under which the optimal derived EO classifier has false-negative rate $\beta$ on both groups.*

The proof of Lemma 1 follows from the fact that the optimal derived classifier can be found by a linear program (Hardt et al., 2016), together with two observations: (i) when minimizing squared-loss subject to EO, a solution can be found at a vertex of the polytope formed by the constraints, and (ii) for some $\mathcal{D}$ and false-positive rates these vertices contain points where $\beta \in \{\beta^0, \beta^1\}$.

We now turn to our first example. In this example, the reason the fairness adjustment worsens EOC is the difference in correlations between classifiers, as described in Section 3.1.

**Example 3** Both classifiers $c_\ell$ have false-negative rates $\beta_\ell^0 = 0.1$ and $\beta_\ell^1 = 0.2$. The correlations are $\rho^0 = 1$ and $\rho^1 = 0.375$ (perhaps both lenders outsource to the same third-party for predictions about group $A = 0$, as in Example 1). Furthermore, suppose that $\mathcal{D}$ and false-positive rates are such that the fairness adjustments lead to classifiers $\tilde{c}_\ell$ with false-negative rates $\tilde{\beta}_\ell^0 = \tilde{\beta}_\ell^1 = 0.1$ (guaranteed to exist by Lemma 1). Such fairness adjustments are accomplished by randomizing each $c_\ell$'s predictions only when $A = 1$ and $c_\ell(X, A) = 0$, and this leads to new correlations $\tilde{\rho}^0 = 1$ and $\tilde{\rho}^1 < 1$.

Observe that, before fairness adjustment, neither classifier is EO, but after fairness adjustment, both classifiers are EO. However, observe also that, before fairness adjustment, the classifiers *are* EOC: First, $\Pr[c_1(X, A) = c_2(X, A) = 0 | Y = 1, A = 0] = 0.1$, because the classifiers are fully correlated on $A = 0$. Second, by (1),

$$\Pr[c_1(X, A) = c_2(X, A) = 0 | Y = 1, A = 1]$$
$$= \rho^1 \sigma_1 \sigma_2 + \beta_1 \beta_2 = 0.375 \cdot 0.2 \cdot 0.8 + 0.2 \cdot 0.2 = 0.1.$$

Finally, observe that after fairness adjustment, the classifiers are no longer EOC. This is because the probability both classifiers misclassify a positive instance in $A = 0$ remains the same, namely,

$\Pr\left[\tilde{c}_1(X, A) = \tilde{c}_2(X, A) = 0 | Y = 1, A = 0\right] = 0.1$. The probability of misclassification in $A = 1$, however, is now different:

$$\Pr\left[\tilde{c}_1(X, A) = \tilde{c}_2(X, A) = 0 | Y = 1, A = 1\right]$$
$$= \tilde{\rho}^1 \cdot 0.1 \cdot 0.9 + 0.1 \cdot 0.1 < 0.1,$$

where the inequality holds since $\tilde{\rho}^1 < 1$.

We now turn to our second example, in which the reason the fairness adjustment worsens EOC is the difference in sets of served borrowers, as described in Section 3.2.

**Example 4** Let $S_1 = \text{supp}(\mathcal{D})$ and $S_2 = \text{supp}(\mathcal{D}|A = 1)$. Note that this implies that $\gamma^0 = 0$ and $\gamma^1 = 1$. Suppose $c_2$ is a perfect classifier, with $\beta_2 = \alpha_2 = 0$. Classifier $c_1$ is a perfect classifier on group $A = 0$ only, and has false-negative rate $\beta_1^0 = 0$ on group $A = 0$ and false-negative rate $\beta_1^1 = \beta > 0$ on group $A = 1$. Note that $c_1$ is not EO, and for $c_2$ the issue is moot since $c_2$ serves only borrowers with $A = 1$. However, note that the classifiers are EOC: a deserving borrower from $A = 1$ will always get a loan offer from $c_2$, and a deserving borrower from $A = 0$ will always get a loan offer from $c_1$.

Now suppose $c_1$ undergoes fairness adjustment to $\tilde{c}_1$ by post-processing, namely, it is derived from $c_1$. Suppose also that false-positive rates and $\mathcal{D}$ are such that, after adjustment, the false-negative rates on both groups are $\tilde{\beta}_1 = \beta$ (guaranteed to exist by Lemma 1). In this case, deserving borrowers from $A = 1$ will still always get a loan offer from $c_2$, but deserving borrowers from $A = 0$ will only get an offer from $\tilde{c}_1$ with probability $1 - \beta$ (and never get an offer from $c_2$). Thus, the classifiers are no longer EOC.

## 5  Experiments

In order to explore the prevalence of our theoretical results we ran several experiments.[3] We describe and report results from the first three here, and the remaining ones in Appendix C. For the first three we used Lending Club loan data for the years 2007-2015, a dataset that includes roughly 890,000 peer-to-peer loans through Lending Club (2021). We used simple classifiers to predict the `loan_status` outcome, and, in particular, whether or not the loan was fully paid, given a set of features that included the loan amount, the interest rate, the number of installments, and the borrower's annual income. As there is no demographic information in the public data, the protected attributes $A$ we used in our experiments were whether or not individuals have a mortgage. We compared the performance in terms of EO and EOC before and after the corresponding fairness adjustment. The fairness adjustment was implemented using the open-source Fairlearn (2025) package.

In the experiments we simulated a situation in which two firms compete over the same pool of borrowers, $S_1 = S_2$, as in Proposition 1. The differences between the firms' classifiers were captured in three different ways in the respective experiments:

**Exp. 1** $c_1$ was a logistic regression and $c_2$ a decision tree, but their training data was identical.

**Exp. 2** Both classifiers were logistic regressions, but their training data was disjoint: one was trained on random examples with a short loan term, and one with a long loan term.

**Exp. 3** $c_1$ was a logistic regression and $c_2$ a decision tree, and their training data was disjoint (as in Exp. 2).

In all three experiments, the training data consisted of a range of $300 - 100,000$ random examples. We ran each experiment with each size of training set 500 times.[4] For each run we calculated the EO of each classifier as well as the EOC level, both before and after the fairness adjustments. The most relevant statistic was then the comparison between the EOC level before fairness adjustments and the EOC level after fairness adjustments. We calculated the percentage of times the EOC level was higher after adjustment than before, as well as standard errors of the mean.

To get a sense of the results, consider the following selected run from experiment 3: Before adjustments, the values were $EO_1 = 0.597192\%$, $EO_2 = 3.430634\%$, and $EOC = \mathbf{0.020774}\%$. Thus,

---

[3]Code is available at `https://github.com/eilamshapira/FairnessUnderCompetition`.

[4]Running all experiments took a few hours on a home computer.

Table 1: 95%-CI for likelihood EOC level increased following fairness adjustment

|        | 300          | 1k           | 3k           | 10k          | 30k          | 100k         |
|--------|--------------|--------------|--------------|--------------|--------------|--------------|
| Exp. 1 | [75.0, 82.2] | [68.0, 76.4] | [55.6, 64.2] | [49.4, 58.2] | [42.0, 50.6] | [26.2, 34.0] |
| Exp. 2 | [75.6, 82.8] | [65.2, 73.8] | [51.8, 60.8] | [35.4, 44.2] | [25.8, 33.8] | [12.6, 19.0] |
| Exp. 3 | [74.2, 81.2] | [63.4, 71.2] | [50.8, 59.6] | [35.6, 43.8] | [27.6, 36.2] | [14.2, 20.6] |

even though both classifiers were somewhat biased, under competition the EOC was nearly 0. However, once the individual classifiers were adjusted for fairness, the values were $\tilde{EO}_1 = 0.403418\%$, $\tilde{EO}_2 = 0.985511\%$, and $\tilde{EOC} = \mathbf{0.444052}\%$. Thus, even though each individual classifier became more fair after adjustment, the result under competition became worse.

To examine the prevalence of this phenomenon we counted the percentage of times the EOC became worse after fairness adjustment. Table 1 reports the results: The rows indicate the experiment number, the columns indicate the size of the training set, and the values in the table are the 95% confidence intervals, in percentages, for the probability that the EOC level was worse after adjustment. We found this to be surprisingly common even for large training sets. For instance, for training sets of size 100k (out of a dataset of size roughly 900k) it occurred 30.1%, 15.8%, and 17.4% of the time in experiments 1, 2, and 3, respectively.

We note that the effect diminishes as the size of the training set increases. The diminishing of the effect for large training sets holds a similar message as the results from Section 3, where we show that the magnitude of the EOC level is roughly the same as that of the false-negative rates. For very large training sets the false-negative rates diminish to 0, and so the levels of EO and EOC diminish in turn.

In addition to examining the prevalence of the effect, we also measured its magnitude. For each run, if the EOC level was higher after fairness adjustment than before, we measured the factor by which the EOC level increased. For each experiment we then calculated the average factor by which the EOC level increased (conditional on increasing), as well as the standard error of the mean. The results are reported in Appendix C due to space constraints. However, to get a sense of the orders-of-magnitude consider the following. For training sets of size 100k (out of a dataset of size roughly 900k), the EOC level increased by a mean factor of 19, 1.3, and 3.1 in experiments 1, 2, and 3, respectively.

In Appendix C we also report on additional experiments we ran. In particular, we ran an experiment with $S_1 \neq S_2$, as in Proposition 4, and simulated the situation described in Example 4. We also ran experiments in which classifiers' training sets were independently chosen, experiments using a different dataset (specifically, Becker and Kohavi, 1996), and experiments with three rather than two competing classifiers. For all experiments, the results were similar to the ones reported here.

## 6  Discussion and future work

Our main insight is that individual fairness is neither necessary nor sufficient for ecosystem fairness. Thus, when making fairness adjustments, their effects on the ecosystem should be considered. An interesting question relates to the kinds of regulations that could lead to ecosystem fairness while also maintaining the benefits of having multiple competing classifiers.[5]

In our analysis we made several simplifying assumptions. First, we focused on EO and EOC. In Appendix B, however, we extend the definitions and analyses to two other common notions of fairness—namely, Equalized Odds and Demographic Parity. In particular, we define variants of these notions for a setting with multiple classifiers, and then show that versions of Propositions 1 and 4 hold for them as well: If the correlations or overlaps in served borrowers between classifiers differ across groups, then classifiers that are fair will not be fair under competition.

Second, for our welfare-based notions of $v$-EO and $v$-EOC, we focused on simple utility functions that satisfy Assumptions 1 or 2. A natural direction that we leave for future work is to consider more general preferences, including ones where deserving borrowers' utilities may differ from others', and where the utility function may depend on any of $(x, a, y)$.

---

[5]Dividing responsibility, for example by allowing only one EO classifier to serve each borrower, would lead to ecosystem fairness, but would not deliver the benefits of having multiple classifiers.

## Acknowledgments and Disclosure of Funding

Research supported by funding from the Israel Science Foundation, award number 2039/24 (Gradwohl), and by funding from the European Research Council (ERC) under the European Union's Horizon 2020 research and innovation programme, grant number 740435 (Tennenholtz).

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

# Appendix

## A  Proofs

**Proof of Proposition 1**  For each $a$, the Pearson correlation between $B_1^a$ and $B_2^a$ is

$$\rho^a = \frac{\Pr\left[B_1^a = B_2^a = 1\right] - \Pr\left[B_1^a = 1\right] \cdot \Pr\left[B_2^a = 1\right]}{\sigma_1 \sigma_2} = \frac{\Pr\left[B_1^a = B_2^a = 1\right] - (1 - \beta_1)(1 - \beta_2)}{\sigma_1 \sigma_2}.$$

Thus,

$$\Pr\left[B_1^a = B_2^a = 1\right] = \rho^a \sigma_1 \sigma_2 + (1 - \beta_1)(1 - \beta_2)$$

and

$$\begin{aligned}
\Pr\left[B_1^a = B_2^a = 0\right] &= 1 - \left(\Pr\left[B_1^a = 1\right] + \Pr\left[B_2^a = 1\right] - \Pr\left[B_1^a = B_2^a = 1\right]\right) \\
&= 1 - (1 - \beta_1) - (1 - \beta_2) + \rho^a \sigma_1 \sigma_2 + (1 - \beta_1)(1 - \beta_2) \\
&= \rho^a \sigma_1 \sigma_2 + \beta_1 \beta_2.
\end{aligned} \tag{1}$$

The level of EOC is $\left|\mathrm{E}\left[d(X, A)|Y = 1, A = 0\right] - \mathrm{E}\left[d(X, A)|Y = 1, A = 1\right]\right|$, where

$$\begin{aligned}
\mathrm{E}\left[d(X, A)|Y = 1, A = a\right] &= 1 - \Pr\left[c_1(X, A) = c_2(X, A) = 0|Y = 1, A = a\right] \\
&= 1 - \Pr\left[B_1^a = B_2^a = 0\right].
\end{aligned}$$

Putting the above together yields

$$\begin{aligned}
&\left|\mathrm{E}\left[d(X, A)|Y = 1, A = 0\right] - \mathrm{E}\left[d(X, A)|Y = 1, A = 1\right]\right| \\
&= \left|\Pr\left[B_1^0 = B_2^0 = 0\right] - \Pr\left[B_1^1 = B_2^1 = 0\right]\right| \\
&= \left|\rho^0 \sigma_1 \sigma_2 + \beta_1 \beta_2 - \rho^1 \sigma_1 \sigma_2 - \beta_1 \beta_2\right| = \sigma_1 \cdot \sigma_2 \cdot \left|\rho^0 - \rho^1\right|,
\end{aligned}$$

as claimed.

Now, the worst case scenario maximizes $\left|\rho^0 - \rho^1\right|$, and so, for example, maximizes $\rho^0$ while minimizing $\rho^1$ (or vice versa). For any $a$, the correlation can be written as $\rho^a = \frac{r^a - \beta_1 \beta_2}{\sigma_1 \sigma_2}$, where $\max\{0, \beta_1 + \beta_2 - 1\} \leq r^a \leq \min\{\beta_1, \beta_2\}$ by the Fréchet-Hoeffding bounds (see, e.g., Nelsen, 2006). Plugging in $r^0 = \min\{\beta_1, \beta_2\}$ and $r^1 = \max\{0, \beta_1 + \beta_2 - 1\}$ yields the result.

To see that this worst case can be achieved, and to obtain an intuitive explanation for the Frèchet-Hoeffding bounds in our setting, recall that

$$r^a = \Pr\left[B_1^a = B_2^a = 1\right] = \Pr\left[c_1(X, A) = c_2(X, A) = 0|Y = 1, A = a\right],$$

the probability that both classifiers misclassify an instance with $Y = 1$. Given that false-negative rates are $\beta_1$ and $\beta_2$, the probability that both classifiers misclassify such an instance is at most $r^0 = \min\{\beta_1, \beta_2\}$, and this occurs when the set of one classifier's misclassified instances are contained in the other's.

The probability that both classifiers misclassify an instance with $Y = 1$ is minimal when the instances on which the classifiers misclassify are maximally disjoint, and in this case the probability that both misclassify an instance is $r^1 = \max\{0, \beta_1 + \beta_2 - 1\}$. In particular, if $\beta_1 + \beta_2 \leq 1$ then the classifiers never misclassify the same positive instance.  ∎

**Proof of Proposition 2**  Recall from the proof of Proposition 1 that

$$\Pr\left[B_1^a = B_2^a = 1\right] = \rho^a \sigma_1 \sigma_2 + (1 - \beta_1)(1 - \beta_2)$$

and

$$\Pr\left[B_1^a = B_2^a = 0\right] = \rho^a \sigma_1 \sigma_2 + \beta_1 \beta_2.$$

Thus, we have that

$$\begin{aligned}
\Pr\left[B_1^a = 1 \cap B_2^a = 0\right] &= \Pr\left[B_1^a = 1\right] - \Pr\left[B_1^a = B_2^a = 1\right] \\
&= 1 - \beta_1 - \rho^a \sigma_1 \sigma_2 - (1 - \beta_1)(1 - \beta_2) \\
&= \beta_2(1 - \beta_1) - \rho^a \sigma_1 \sigma_2
\end{aligned}$$

and, analogously,
$$\Pr\left[B_1^a = 0 \cap B_2^a = 1\right] = \beta_1(1 - \beta_2) - \rho^a \sigma_1 \sigma_2.$$

Thus, for each $a$,
$$\begin{aligned}
\mathrm{E}\left[v(X, A, Y, R(X, A))|Y = 1, A = a\right] &= k \cdot \Pr\left[c_1(X, A) = c_2(X, A) = 1|Y = 1, A = a\right] \\
&\quad + \Pr\left[c_1(X, A) \neq c_2(X, A)|Y = 1, A = a\right] \\
&= k \cdot \Pr\left[B_1^a = B_2^a = 1\right] + \Pr\left[B_1^a \neq B_2^a\right] \\
&= k\rho^a \sigma_1 \sigma_2 + k(1 - \beta_1)(1 - \beta_2) + \beta_2(1 - \beta_1) + \beta_1(1 - \beta_2) - 2\rho^a \sigma_1 \sigma_2 \\
&= (k - 2)\rho^a \sigma_1 \sigma_2 + k - (k - 1)(\beta_1 + \beta_2 - \beta_1 \beta_2).
\end{aligned}$$

The level of $v$-EOC is thus
$$\begin{aligned}
&\left|\mathrm{E}\left[v(X, A, Y, R(X, A))|Y = 1, A = 0\right] - \mathrm{E}\left[v(X, A, Y, R(X, A))|Y = 1, A = 1\right]\right| \\
&= \left|(k - 2)\rho^0 \sigma_1 \sigma_2 - (k - 2)\rho^1 \sigma_1 \sigma_2\right| \\
&= \sigma_1 \sigma_2 \left|(k - 2)(\rho^0 - \rho^1)\right|.
\end{aligned}$$

As in the proof of Proposition 1, the worst case scenario maximizes $\left|\rho^0 - \rho^1\right|$, and so, for example, maximizes $\rho^0$ while minimizing $\rho^1$ (or vice versa). Again, as before, the worst-case difference is $\frac{1}{\sigma_1 \sigma_2}\left(\min\{\beta_1, \beta_2\} - \max\{0, \beta_1 + \beta_2 - 1\}\right)$, which leads to the claimed worst case level of $v$-EOC. ∎

**Proof of Proposition 3** The worst case occurs when the probability that all classifiers misclassify positive instances is minimal in group $A = 1$ and maximal in group $A = 0$ (or vice versa). To maximize $\Pr\left[c_1(X, A) = \ldots = c_n(X, A) = 0|A = 0, Y = 1\right]$ the overlap in instances on which each $c_i$ misclassifies has to be maximal, which occurs when the instances are contained in one anther. Formally, if $\beta_1 \leq \ldots \leq \beta_n$, then
$$c_n(x, 0) = 1 \Rightarrow \ldots \Rightarrow c_1(x, 0) = 1.$$

In this case,
$$\Pr\left[c_1(X, A) = \ldots = c_n(X, A) = 0|A = 0, Y = 1\right] = \beta_n = \min_{i \in L} \beta_i.$$

To minimize $\Pr\left[c_1(X, A) = \ldots = c_n(X, A) = 0|A = 1, Y = 1\right]$ the overlap in instances on which each $c_i$ misclassifies has to be minimal, which occurs when the instances are maximally disjoint. Equivalently, the overlap in positive instances in which the $c_i$'s correctly classify is maximally disjoint. If $\sum_j(1 - \beta_j) \geq 1$ then
$$\{(x, 1, 1) : c_1(x, 1) = \ldots = c_n(x, 1) = 0\} = \emptyset.$$

Otherwise, in the worst case each classifier $c_i$ correctly classifies a unique set of $1 - \beta_i$ positive instances, and in this case
$$\Pr\left[c_1(X, A) = \ldots = c_n(X, A) = 0|A = 0, Y = 1\right] = \sum_{j \in L} \beta_j - 1.$$

Thus, the worst-case level of EOC is $\min_{i \in L} \beta_i - \max\left\{0, \sum_{j \in L} \beta_j - 1\right\}$.

Finally, if $\beta_i = \beta \leq 1 - 1/n$ for all $i$ then $\min_{i \in L} \beta_i = \beta$ and $\max\left\{0, \sum_{j \in L} \beta_j - 1\right\} = 0$. ∎

**Proof of Proposition 4** For any $a \in A$,
$$\begin{aligned}
\mathrm{E}\left[d(X, A)|Y = 1, A = a\right] &= 1 - \Pr\left[c_1(X, A) = c_2(X, A) = 0|Y = 1, A = a\right] \\
&= 1 - \Big(\Pr\left[c_1(X, A) = 0 \cap (X, A, Y) \in S^1 \setminus S^2 \mid Y = 1, A = a\right] \\
&\quad + \Pr\left[c_2(X, A) = 0 \cap (X, A, Y) \in S^2 \setminus S^1 \mid Y = 1, A = a\right] \\
&\quad + \Pr\left[c_1(X, A) = c_2(X, A) = 0 \cap (X, A, Y) \in S^1 \cap S^2 \mid Y = 1, A = a\right]\Big) \\
&= 1 - \left(\left(\gamma_1^a - \gamma^a\right)\beta_1 + \left(\gamma_2^a - \gamma^a\right)\beta_2 + \gamma^a \beta_1 \beta_2\right). \\
&= 1 - \left(\left(1 - \gamma_2^a\right)\beta_1 + \left(1 - \gamma_1^a\right)\beta_2 + \gamma^a \beta_1 \beta_2\right).
\end{aligned}$$

The level of EOC is thus

$$\left| \mathrm{E}\left[ d(X, A) | Y = 1, A = 0 \right] - \mathrm{E}\left[ d(X, A) | Y = 1, A = 1 \right] \right|$$
$$= \left| (\gamma_2^0 - \gamma_2^1)\beta_1 + (\gamma_1^0 - \gamma_1^1)\beta_2 + (\gamma^1 - \gamma^0)\beta_1\beta_2 \right|.$$

The worst case is constructed as follows. Classifier $\ell = \arg\max_i \beta_i$ serves all borrowers from group $A = 0$, and the other classifier $(3 - \ell)$ does not serve any borrower from this group. Additionally, both classifiers serve all borrowers with $A = 1$. Formally, $\gamma_\ell^0 = \gamma_\ell^1 = 1$ and $\gamma_{3-\ell}^0 = 0$ and $\gamma_{3-\ell}^1 = 1$. This implies also that $\gamma^0 = 0$ and $\gamma^1 = 1$, and so yields a level of EOC equal to $\max\{\beta_1, \beta_2\} - \beta_1\beta_2$.

To see that this is maximal, observe that

$$\gamma_2^0 - \gamma_2^1 + \gamma_1^0 - \gamma_1^1 = \gamma_2^0 + \gamma_1^0 - (\gamma_2^1 - \gamma_1^1) = 1 + \gamma^0 - (1 + \gamma^1) = \gamma^0 - \gamma^1.$$

Thus, $(\gamma_2^0 - \gamma_2^1)\beta_1 + (\gamma_1^0 - \gamma_1^1)\beta_2 \le (\gamma^0 - \gamma^1) \cdot \max\{\beta_1, \beta_2\}$, and so the level of EOC is at most

$$(\gamma^0 - \gamma^1)(\max\{\beta_1, \beta_2\} - \beta_1\beta_2) \le \max\{\beta_1, \beta_2\} - \beta_1\beta_2.$$

∎

**Proof of Lemma 1** For a classifier $c$ and a group $a \in A$, let $\lambda^a(c) = (\alpha^a, 1 - \beta^a)$, where $\alpha^a = \Pr\left[ c_L(X, A, Y) = 1 | Y = 0, A = a \right]$ is the false-positive rate of the classifier in group $a$, and $1 - \beta^a = \Pr\left[ c_L(X, A, Y) = 1 | Y = 1, A = a \right]$ is the true-positive rate of the classifier in group $a$. Hardt et al. (2016) show that any derived classifier $\tilde{c}$ satisfies $\lambda^a(\tilde{c}) \in \mathrm{convhull}\left\{ (0,0), \lambda^a(c), \lambda^a(1 - c), (1,1) \right\}$, where $(1 - c)$ is the classifier $c$ but with predictions flipped. They then show that the optimal derived classifier can be found using the following linear program:

$$\min_{\tilde{c}} \quad \mathrm{E}\left[ \ell(\tilde{c}, c) \right]$$
$$\text{s.t.} \quad \lambda^a(\tilde{c}) \in \mathrm{convhull}\left\{ (0,0), \lambda^a(c), \lambda^a(1 - c), (1,1) \right\}, \quad \forall a \in A$$
$$\lambda_2^0(\tilde{c}) = \lambda_2^1(\tilde{c})$$

The second constraint above states that the true-positive rates of the classifier are equal on both $a \in A$, and so the classifier $\tilde{c}$ must be EO. The optimization problem is a squared-loss minimization.

Let us fix the distribution $\mathcal{D}$ to be uniform over all of $X \times Y \times A$. Thus, for each group $a$ and label $y \in \{0, 1\}$, we have that $\Pr\left[ A = a \right] = \Pr\left[ Y = y \right] = 1/2$. We also assume, without loss of generality, that $\beta^0 > \beta^1$.

The optimal derived classifier $\tilde{c}$ is then one of the following:

1. Let $\tilde{c} \equiv c$ except that, on some fraction of instances with $c(x, 0) = 0$ fix $\tilde{c}(x, 0) = 1$, so that
$$\lambda^0(\tilde{c}) = (1 - \delta^0)\lambda^0(c) + \delta^0 \cdot (1,1) = (\tilde{\alpha}^0, 1 - \beta^1)$$
for $\delta^0 = (\beta^0 - \beta^1)/\beta^0$ and some $\tilde{\alpha}^0$. In this case, $\tilde{\beta}^0 = \beta^1$. The cost of this in terms of squared loss is the fraction of instances where $c(x, 0) = 0$ that were flipped to $\tilde{c}(x, 0) = 1$, namely, $\delta^0(1 - \alpha^0 + \beta^0)/4$.

2. Let $\tilde{c} \equiv c$ except that, on some fraction of instances with $c(x, 1) = 1$ fix $\tilde{c}(x, 1) = 0$, so that
$$\lambda^1(\tilde{c}) = (1 - \delta^1)\lambda^1(c) + \delta^1 \cdot (0,0) = (\tilde{\alpha}^1, 1 - \beta^0)$$
for $\delta^1 = (\beta^0 - \beta^1)/(1 - \beta^1)$ and some $\tilde{\alpha}^1$. In this case, $\tilde{\beta}^1 = \beta^0$. The cost of this in terms of squared loss is the fraction of instances where $c(x, 1) = 1$ that were flipped to $\tilde{c}(x, 1) = 0$, namely, $\delta^1(\alpha^1 + 1 - \beta^0)/4$.

To see that one of the above is an optimal derived classifier, note first that the optimal solution cannot be in the strict interior of the polytope. Furthermore, if $(1 - \alpha^0 + \beta^0)/\beta^0 \ne (\alpha^1 + 1 - \beta^0)/(1 - \beta^1)$ then considering a mixture of (1) and (2)—on some fraction of instances with $c(x, 0) = 0$ setting $\tilde{c}(x, 0) = 1$ and on some fraction of instances with $c(x, 1) = 1$ setting $\tilde{c}(x, 1) = 0$—is suboptimal. Finally, note that setting some fraction of instances with $c(x, 0) = 1$ to $\tilde{c}(x, 0) = 0$ only increases the EO, as does setting some fraction of instances with $c(x, 1) = 0$ to $\tilde{c}(x, 1) = 1$, and so cannot be part of an optimal classifier.

Finally, an optimal derived EO classifier in which the false-negative rate is $\beta^1$ (resp., $\beta^0$) on both groups is one where false-positive rates satisfy $(1 - \alpha^0 + \beta^0)/\beta^0 < (\alpha^1 + 1 - \beta^0)/(1 - \beta^1)$ (resp., $(1 - \alpha^0 + \beta^0)/\beta^0 > (\alpha^1 + 1 - \beta^0)/(1 - \beta^1)$). Under equality, both classifiers are optimal. ∎

# B  Additional fairness notions

In this section we discuss extensions of our results to two additional, common notions of fairness. We state the two notions for the setting of a single classifier, extend the definitions to multiple competing classifiers, and then show that versions of Proposition 1 and 4 hold for these notions as well: If the correlations or overlaps in served borrowers between classifiers differ across groups, then classifiers that are fair will not be fair under competition.

The first notion we consider is Equalized Odds (ED), a strengthening of EO, which requires both the classifier's false-negative and false-positive rates to be equal across groups (Hardt et al., 2016). The second we consider is Demographic Parity (DP), which requires the total fraction of approved borrowers to be equal across groups (Agarwal et al., 2018; Dwork et al., 2012).

ED: The ED level of $c_\ell$ is

$$\max_{y\in\{0,1\}} \left\{ \left| E\left[c_\ell(X, A)|Y = y, A = 0\right] - E\left[c_\ell(X, A)|Y = y, A = 1\right] \right| \right\}.$$

A classifier is ED if its ED level is 0.

DP: The DP level of $c_\ell$ is $\left| E\left[c_\ell(X, A)|A = 0\right] - E\left[c_\ell(X, A)|A = 1\right] \right|$. A classifier is DP if its DP level is 0.

We now generalize the above definitions of fairness to a setting with multiple competing lenders, ED under competition (EDC) and DP under competition (DPC). Fix classifiers $c = (c_1, \ldots, c_{|L|})$, and denote by $R(x, a) = \sum_{\ell\in L} c_\ell(x, a)$ the number offers made to a borrower with observables $(x, a)$. Also, denote by $d(x, a) = \Pr\left[R(x, a) \geq 1\right]$, the probability that at least one lender offers a borrower with observables $(x, a)$ a loan.

EDC: The EDC level of classifiers $c$ is

$$\max_{y\in\{0,1\}} \left\{ \left| E\left[d(X, A)|Y = y, A = 0\right] - E\left[d(X, A)|Y = y, A = 1\right] \right| \right\}.$$

Classifiers are EDC if their EDC level is 0.

DPC: The DPC level of classifiers $c$ is $\left| E\left[d(X, A)|A = 0\right] - E\left[d(X, A)|A = 1\right] \right|$. Classifiers are DPC if their DPC level is 0.

## B.1  Equalized Odds under Competition

In this section we show that classifiers that are ED need not be EDC. Consider the following simple observation, which follows from the fact that the definition of ED is strictly stronger than that of EO:

**Observation 1** *Fix classifiers* $c = (c_1, \ldots, c_L)$.

- *If $c_\ell$ is ED, then it is also EO.*

- *The EDC level of classifiers $c$ is at least their EOC level.*

Fix two ED classifiers $c_1$ and $c_2$, and define $\beta_\ell$, $\sigma_\ell$, and $\rho^a$ as in Section 3.1. Observation 1 implies the following corollary of Proposition 1.

**Corollary 4** *For two ED classifiers $c_1$ and $c_2$ with false-negative rates $\beta_1$ and $\beta_2$, the level of EDC is at least $\sigma_1 \cdot \sigma_2 \cdot \left|\rho^0 - \rho^1\right|$. In the worst case, the level of EDC is at least $\min\{\beta_1, \beta_2\} - \max\{0, \beta_1 + \beta_2 - 1\}$.*

Similarly, fix $S_\ell$, $\gamma_\ell^a$, and $\gamma^a$ as in Section 3.2. Observation 1 implies the following corollary of Proposition 4.

**Corollary 5** *For two uncorrelated ED classifiers $c_1$ and $c_2$ with false-negative rates $\beta_1$ and $\beta_2$, the level of EDC is at least*

$$\left|(\gamma_2^0 - \gamma_2^1)\beta_1 + (\gamma_1^0 - \gamma_1^1)\beta_2 + (\gamma^1 - \gamma^0)\beta_1\beta_2\right|.$$

*In the worst case, the level of EDC is at least $\max\{\beta_1, \beta_2\} - \beta_1\beta_2$.*

## B.2 Demographic Parity under Competition

In this section we show that classifiers that are DP need not be DPC.

### B.2.1 Correlations between classifiers

Fix two DP classifiers $c_1$ and $c_2$, and denote their approval probabilities by $\eta_\ell = \Pr\left[c_\ell(X, A) = 1\right]$. We define two Bernoulli random variables that capture the probability of approval on the two groups $a \in A$: For each $a \in A$ and each $\ell \in \{1, 2\}$, let the Bernoulli random variable

$$C_\ell^a \equiv c_\ell(X, A)\,|_{(A=a)}.$$

Each $C_\ell^a$ is the output of classifier $c_\ell$ on random instances with $A = a$, and so $\mathrm{E}\left[C_\ell^a\right] = \eta_\ell$. Denote by $\sigma_\ell = \sqrt{\eta_\ell(1 - \eta_\ell)}$ the standard deviation of $C_\ell^0$, and note that, because $c_\ell$ is DP, $\sigma_\ell$ is also the standard deviation of $C_\ell^1$. Finally, for each $a$ let $\rho^a$ denote the Pearson correlation coefficient between $C_1^a$ and $C_2^a$. The Pearson correlation captures the extent to which the approvals of the classifiers correlate with one another.

We now quantify the DPC level of two classifiers using these correlation coefficients, and determine the worst case.

**Proposition 5** *For two DP classifiers $c_1$ and $c_2$ with approval probabilities $\eta_1$ and $\eta_2$, the level of DPC is $\sigma_1 \cdot \sigma_2 \cdot \left|\rho^0 - \rho^1\right|$. In the worst case, the level of DPC is $\min\{1 - \eta_1, 1 - \eta_2\} - \max\{0, 1 - \eta_1 - \eta_2\}$.*

The proof of Proposition 5 is nearly identical to that of Proposition 1, with $C_\ell^a$ replacing $B_\ell^a$, $1 - \eta_\ell$ replacing $\beta_\ell$, and without conditioning any of the probabilities and expectations on $Y = 1$.

Proposition 5 yields the following simple corollary.

**Corollary 6** *Suppose $\eta = \eta_1 = \eta_2$. If $\eta \geq 1/2$, then the worst-case level of DPC is $1 - \eta$. If $\eta \leq 1/2$, then the worst-case level of DPC is $\eta$.*

Recall that, under EOC, the more accurate EO classifiers get, the lower the worst-case level of EOC. With DPC this also holds. To see this, observe that as DP classifiers get more accurate—specifically, as their false-positive and false-negative rates approach 0—the correlations $\rho^0$ and $\rho^1$ between the classifiers also approach 0. By Proposition 5, this implies that the level of DPC approaches 0.

### B.2.2 Different sets of borrowers

Fix two DP classifiers $c_1$ and $c_2$ with approval probabilities $\eta_1$ and $\eta_2$, and let $S_\ell$, $\gamma_\ell^a$, and $\gamma^a$ be as in Section 3.2. Consider the following variant of Definition 1, which essentially removes the conditioning on $Y = 1$:

**Definition 2** *DP classifiers $c_1$ and $c_2$ with approval probabilities $\eta_1$ and $\eta_2$ are* uncorrelated *if (i) for each $\ell$,*

$$\Pr\left[c_\ell(X, A) = 1 | (X, A, Y) \in S_1 \cap S_2\right]$$
$$= \Pr\left[c_\ell(X, A) = 1 | (X, A, Y) \in S_\ell \setminus S_{3-\ell}\right],$$

*and (ii) for every $(x, y, a) \in S_1 \cap S_2$ it holds that $\Pr\left[c_1(x, a) = c_2(x, a) = 1\right] = \eta_1 \cdot \eta_2$.*

**Proposition 6** *For two uncorrelated DP classifiers $c_1$ and $c_2$ with approval probabilities $\eta_1$ and $\eta_2$, the level of DPC is*

$$\left|(\gamma_2^0 - \gamma_2^1)(1 - \eta_1) + (\gamma_1^0 - \gamma_1^1)(1 - \eta_2) + (\gamma^1 - \gamma^0)(1 - \eta_1)(1 - \eta_2)\right|.$$

*In the worst case, the level of EOC is $\max\{1 - \eta_1, 1 - \eta_2\} - (1 - \eta_1)(1 - \eta_2)$.*

The proof of Proposition 6 is nearly identical to that of Proposition 4, with $1 - \eta_\ell$ replacing $\beta_\ell$ and without conditioning any of the probabilities and expectations on $Y = 1$.

Proposition 6 yields the following simple corollary.

**Corollary 7** *For two uncorrelated DP classifiers $c_1$ and $c_2$ with approval probabilities $\eta_1 = \eta_2 = \eta$, the level of DPC is $\eta \cdot (1 - \eta) \cdot \left|\gamma^0 - \gamma^1\right|$.*

# C Additional experiments

In this section we describe and report the results of additional experiments that we ran.

## C.1 Effect size

In addition to examining the prevalence of the effect of fairness adjustment on the EOC, we also measured its magnitude. For each run in Experiments 1-3, if the EOC level was higher after fairness adjustment than before, we measured the factor by which the EOC level increased. For each experiment we then calculated the average factor by which the EOC level increased (conditional on increasing), as well as the standard error of the mean. Table 2 reports the results: The values in the table are the 95% confidence intervals for the factor by which the EOC level increased after adjustment (conditional on increasing).

Table 2: 95%-CI for effect size (ratio > 1 only)

|  | 300 | 1k | 3k | 10k | 30k | 100k |
|---|---|---|---|---|---|---|
| Exp. 1 | [10.8, 30.8] | [7.8, 16.3] | [7.6, 121.0] | [6.0, 19.9] | [7.3, 23.0] | [7.5, 31.0] |
| Exp. 2 | [21.1, 111.8] | [8.3, 30.1] | [2.9, 9.4] | [1.8, 2.2] | [1.6, 1.8] | [1.3, 1.4] |
| Exp. 3 | [14.3, 27.2] | [10.0, 74.2] | [8.2, 16.7] | [4.8, 32.4] | [2.5, 3.6] | [1.9, 4.3] |

## C.2 Experiment with different subsets

In this experiment we simulated a situation in which $S_1 \neq S_2$, as in Proposition 4. In particular, we simulated the situation described in Example 4, where $S_1 = \mathrm{supp}(\mathcal{D})$ and $S_2 = \mathrm{supp}(\mathcal{D}|A = 1)$, and where $A = 1$ is the set of individuals with a mortgage. Classifier $c_1$ was trained on a random set of examples taken from the entire dataset, whereas classifier $c_2$ was trained on a random set of examples consisting only of individuals with a mortgage. Both classifiers were logistic regressions, but since only classifier $c_1$ served individuals with both values of $A$, only that classifier underwent fairness adjustment. As in the previous experiments, we measured the level of EOC before and after the fairness adjustment.

Our results for experiment are reported in Tables 3 and 4, and are the following: For training sets of size up 10k, fairness adjustment leads to worse EOC in a significant fraction of runs. For larger training sets, however, the effect disappears. This is in line with the previous experiments, except that the effect diminishes for smaller training sets. Even when the effect is prevalent, however, its size is rather small.

Table 3: 95%-CI for likelihood EOC level increased following fairness adjustment

|  | 300 | 1k | 3k | 10k | 30k | 100k |
|---|---|---|---|---|---|---|
| Exp. 4 | [52.2, 60.4] | [34.2, 42.4] | [17.8, 25.2] | [5.4, 10.0] | [0.8, 3.2] | [0.0, 0.0] |

Table 4: 95%-CI for effect size (ratio > 1 only)

|  | 300 | 1k | 3k | 10k | 30k | 100k |
|---|---|---|---|---|---|---|
| Exp. 4 | [1.5, 1.8] | [1.2, 1.3] | [1.1, 1.2] | [1.1, 1.1] | [1.0, 1.1] | [N/A] |

## C.3 Independent samples

We ran two experiments that were similar to Experiments 1-3, except that the training data was chosen independently for each classifier.

**Exp. 5** Both classifiers were decision trees, and their training data was sampled independently.

**Exp. 6** Classifier $c_1$ was a logistic regression and $c_2$ a decision tree, and their training data was sampled independently.

The results are reported in Tables 5 and 6.

Table 5: 95%-CI for harm likelihood following fairness adjustment

|        | 300          | 1k           | 3k           | 10k          | 30k          | 100k         |
|--------|--------------|--------------|--------------|--------------|--------------|--------------|
| Exp. 5 | [56.2, 65.0] | [57.0, 65.6] | [50.2, 58.8] | [51.0, 59.6] | [41.2, 49.6] | [24.0, 31.4] |
| Exp. 6 | [71.8, 79.2] | [67.0, 74.4] | [55.0, 63.6] | [49.6, 58.4] | [45.0, 54.4] | [28.0, 36.0] |

Table 6: 95%-CI for effect size (ratio > 1 only)

|        | 300           | 1k          | 3k          | 10k          | 30k         | 100k        |
|--------|---------------|-------------|-------------|--------------|-------------|-------------|
| Exp. 5 | [5.4, 9.4]    | [6.9, 14.0] | [6.4, 44.8] | [7.3, 14.1]  | [5.7, 18.5] | [3.2, 18.7] |
| Exp. 6 | [10.7, 858.3] | [8.3, 24.5] | [5.8, 10.3] | [11.3, 35.8] | [5.6, 11.4] | [5.7, 12.9] |

## C.4 Different dataset

We ran Experiments 1-3 on a different dataset from the Lending Club loan data—the Adult Census Income dataset from the UC Irvine Machine Learning Repository (Becker and Kohavi, 1996). This dataset contains data on roughly 40,000 individuals. We used simple classifiers to predict whether an individual's income is above \$50,000, given a set of features that included age, education, work hours per week, occupation, and relationship status. The protected attribute $A$ was gender.

**Exp. 1'** Classifier $c_1$ was a logistic regression and $c_2$ a decision tree, but their training data was identical.

**Exp. 2'** Both classifiers were logistic regressions, but their training data was disjoint: one was trained on random examples of White individuals, and one on non-White individuals.

**Exp. 3'** Classifier $c_1$ was a logistic regression and $c_2$ a decision tree, and their training data was disjoint (as in Exp. 2).

The results are reported in Tables 7 and 8.

Table 7: 95%-CI for harm likelihood following fairness adjustment

|         | 300          | 1k           | 3k           | 10k          |
|---------|--------------|--------------|--------------|--------------|
| Exp. 1' | [43.8, 52.6] | [29.6, 37.4] | [13.4, 20.2] | [2.0, 5.2]   |
| Exp. 2' | [52.8, 63.1] | [45.4, 54.4] | [13.6, 20.2] | [0.0, 1.2]   |
| Exp. 3' | [43.2, 52.8] | [39.4, 48.1] | [34.4, 43.0] | [18.4, 25.4] |

Table 8: 95%-CI for effect size (ratio > 1 only)

|         | 300         | 1k          | 3k          | 10k        |
|---------|-------------|-------------|-------------|------------|
| Exp. 1' | [3.9, 11.8] | [4.4, 8.8]  | [2.2, 14.2] | [1.1, 1.3] |
| Exp. 2' | [4.7, 7.9]  | [4.7, 12.1] | [1.8, 2.5]  | [1.0, 2.0] |
| Exp. 3' | [3.1, 26.2] | [4.4, 8.2]  | [6.1, 51.7] | [2.3, 3.7] |

## C.5 More than two classifiers

We ran several experiments that included three, rather than two, classifiers. As before, we measured the level of EOC before and after all three underwent a fairness adjustment. In particular, we ran the following three experiments on the Lending Club data:

**Exp. 7** Classifier $c_1$ was a logistic regression, $c_2$ a decision tree, and $c_3$ a random forest, and their training data was identical.

**Exp. 8** All classifiers were logistic regressions, and their training data was independently sampled.

**Exp. 9** Classifier $c_1$ was a logistic regression, $c_2$ a decision tree, and $c_3$ a random forest, and their training data was independently sampled.

Tables 9 and 10 report the results.

Table 9: 95%-CI for harm likelihood following fairness adjustment

|        | 300          | 1k           | 3k           | 10k          | 30k          | 100k         |
|--------|--------------|--------------|--------------|--------------|--------------|--------------|
| Exp. 7 | [74.2, 81.2] | [71.2, 78.6] | [60.0, 68.2] | [49.6, 58.2] | [44.6, 53.4] | [28.4, 36.4] |
| Exp. 8 | [80.4, 87.0] | [79.0, 85.8] | [77.4, 84.2] | [65.4, 73.0] | [54.6, 63.4] | [30.8, 39.0] |
| Exp. 9 | [74.2, 81.4] | [73.0, 80.8] | [65.0, 73.2] | [63.6, 71.6] | [52.0, 60.8] | [34.2, 43.2] |

Table 10: 95%-CI for effect size (ratio $> 1$ only)

|        | 300           | 1k           | 3k          | 10k         | 30k         | 100k        |
|--------|---------------|--------------|-------------|-------------|-------------|-------------|
| Exp. 7 | [11.7, 57.7]  | [8.8, 19.1]  | [7.5, 19.7] | [6.2, 22.8] | [5.5, 11.6] | [5.4, 22.7] |
| Exp. 8 | [25.2, 133.7] | [12.4, 32.5] | [6.0, 10.9] | [3.2, 3.7]  | [2.2, 2.5]  | [1.5, 1.7]  |
| Exp. 9 | [17.1, 106.7] | [13.6, 26.3] | [9.6, 28.9] | [7.6, 16.1] | [7.2, 15.2] | [5.9, 14.3] |

In addition, we ran Experiments 7', 8', and 9', which were the same as 7, 8, and 9 except that they used the Adult Census Income dataset. The results are reported in Tables 11 and 12.

Table 11: 95%-CI for harm likelihood following fairness adjustment

|         | 300          | 1k           | 3k            | 10k         | 30k        |
|---------|--------------|--------------|---------------|-------------|------------|
| Exp. 7' | [41.2, 50.2] | [31.2, 39.0] | [9.8, 15.6]   | [0.4, 2.2]  | [0.0, 0.0] |
| Exp. 8' | [49.2, 57.8] | [30.2, 38.0] | [7.8, 12.8]   | [0.0, 0.6]  | [0.0, 0.0] |
| Exp. 9' | [47.8, 56.4] | [34.6, 43.2] | [16.6, 23.6]  | [5.2, 10.0] | [0.0, 1.0] |

Table 12: 95%-CI for effect size (ratio $> 1$ only)

|         | 300          | 1k          | 3k         | 10k        | 30k        |
|---------|--------------|-------------|------------|------------|------------|
| Exp. 7' | [2.7, 3.8]   | [2.9, 8.6]  | [1.3, 2.0] | [1.1, 1.7] | [N/A]      |
| Exp. 8' | [5.7, 53.2]  | [3.5, 7.7]  | [1.5, 2.5] | [1.0, 1.0] | [N/A]      |
| Exp. 9' | [3.8, 15.2]  | [3.4, 12.7] | [1.9, 3.4] | [1.1, 1.2] | [1.0, 1.1] |

