# OpenReview forum: "Fairness under Competition"
_NeurIPS.cc/2025/Conference — NeurIPS 2025 poster_

### Official Review · Reviewer_7AYg · 2025-06-26

**Clarity:** 3
**Significance:** 3
**Originality:** 3
**Rating:** 5
**Confidence:** 4

**Summary:**

This work proposes a novel concept of *fairness under competition*, where the goal is not only to study the *fairness of individual predictive models* but to analyze the overall fairness of the *deployment ecosystem* where different competing decision-makers may own such varying predictive models. The authors identify two primary sources of ecosystem unfairness in the presence of multiple predictive models: i) the correlation of the prediction scores between the classifiers, and ii) the data overlap between the different models in deployment. Using illustrative theoretical examples, the authors further show how certain fairness modifications on the predictive models can improve the fairness of *individual predictors* but deteriorate the *ecosystem's overall fairness*. The authors conduct experiments on one real-world dataset to show how different ecosystem setups and varying data sizes can possibly impact the fairness under competition.

**Questions:**

1. Are there any fairness notions that the general welfare formulation cannot capture?
2. Can the authors comment on the potential relationship of the different decision-makers using different subsamples and selection labeling? If the two subsamples are i.i.d., is there still a chance of having ecosystem unfairness stemming from it?

**Ethical Concerns:**

["NO or VERY MINOR ethics concerns only"]

**Final Justification:**

The authors proposed a novel paradigm to study fairness where the focus is not a single predictive or decision-making model but an entire ecosystem, as is the case in most real-world settings. This paradigm can open new directions to pursue in algorithmic fairness. Through the rebuttal, the authors also conducted additional evaluations that further strengthen their claims and provide more empirical grounding.

**Limitations:**

Yes

**Quality:**

3

**Strengths And Weaknesses:**

## Strenghts

- This work proposes a novel perspective on the well-established research space of algorithmic fairness. While most works focus on studying the fairness of individual predictive models, this work provides a novel approach to studying the fairness of the entire market ecosystem.
- The theoretical framework helps to identify two critical components that can affect ecosystem unfairness. This aspect is important in improving the practical applicability of these new notions and informing potential mitigation strategies.
- The authors provide some theoretical analysis in the appendix regarding how multiple classifiers in an ecosystem can impact the ecosystem's unfairness.


## Weaknesses

- The evaluations are limited, focusing on only one dataset and very specific subsamples from the underlying data distribution. A more thorough evaluation would have helped. Specifically, it would also be interesting to study the effects across multiple datasets, in the presence of multiple classifiers, and for different subsampling of data by each of the classifiers.
- The factors discussed by the authors that impact ecosystem unfairness are also related to other well-studied aspects like selective labeling, predictive multiplicity, etc. For instance, the gap in overall fairness from different subsampling of data is very related to the issue of the selective labeling problem and how that interacts with fairness issues.
- The analyses are conducted using only one fairness notion (equal opportunity). While this notion is very relevant to the lending scenario explained in the paper, it would be helpful to understand further ecosystem unfairness from the lens of other fairness notions and their empirical evaluations.
- The evaluations reported the likelihood of individual fairness interventions worsening the overall ecosystem unfairness; however, one cannot infer the magnitude of the worsening of the ecosystem unfairness.

---

> ### Author Rebuttal · Authors · 2025-07-29
>
> Thank you for all your comments and suggestions!
>
> In the following, we first address your questions, and then also your comments from the "Strengths and Weaknesses" section.
>
> ### Questions
>
> 1. The general welfare formulation captures existing notions of group fairness, but cannot capture notions of individual fairness.
>
> 2. The possible connection to selection labeling is interesting, but as of now we are not able to draw a clear analogy. However, we did run our experiment with two i.i.d. subsamples, and the results were very similar to the experiments in the paper in terms of ecosystem unfairness. More specifically, we ran two additional experiments – in ExpA there were two identical classifiers (decision trees), using iid data, and in ExpB there were two different classifiers (decision tree and logistic regression), using iid data. The results were the following. (We report only sample size 30k for brevity, with 95% CIs and also effect size. The effect size is described below, in the reply to the comment starting with "The evaluations reported...").
>
>     > ExpA: 45.4% [41.2%, 49.6%]  | effect size: 10.90x [5.73, 18.53]
>     >
>     > ExpB: 49.8% [45.0%, 54.4%]  | effect size: 8.06x [5.56, 11.35]
>
>     We plan to include the full set of new experimental results in the paper.
>
>
> ### Other comments
>
> **Comment starting with "The evaluations are limited...":** Following this suggestion (also made by another reviewer), we ran the same experiments on a different dataset: The Adult Income dataset from the UC Irvine Machine Learning Repository. The dataset is smaller, and contains 48,000 instances. We ran our experiments to predict whether or not income is above \$50,000, using age, education level, work hours per week, occupation, and family status as features. The protected attribute was gender. The results were broadly similar to our experiment, in terms of orders-of-magnitude. For brevity, we report here the 95%-CIs for a sample size of 3000:
>
> > Exp1:  [13.4, 20.2]
> >
> > Exp2:  [34.4, 43.0]
> >
> > Exp3:  [45.4, 54.4]
> >
> > Exp4:  [61.8, 69.8]
>
>
> **Comment starting with "The analyses are conducted using only one fairness notion...":** We agree. However, the other relevant fairness notions we considered also introduced additional subtleties, which we felt crowded out the main message of the current paper. It is certainly a direction we are pursuing in subsequent work.
>
> **Comment starting with "The evaluations reported...":** Following this comment, we ran the experiments again, this time recording the magnitudes of the effect. Specifically, for each experiment and sample size, we look at all the runs where the ecosystem fairness worsened, calculated the factor by which it worsened, and took an average of these factors. The results for sample size 30k were the following, with 95% CIs. (Other sizes were similar, reporting here only one size for brevity.)
>
> > Exp1: 13.82x [7.25, 22.98]
> >
> > Exp2: 1.66x [1.57, 1.76]
> >
> > Exp3: 3.05x [2.54, 3.62]
> >
> > Exp4: 1.06x [1.03, 1.08]
>
> Interestingly, in Exp1 the level of unfairness grew by an order or magnitude, in Exp2 and Exp3 it grew by a small factor, and in Exp4 it barely grew.

---

> > ### Comment · Reviewer_7AYg · 2025-08-04
> >
> > I thank the authors for their thoughtful rebuttal and the additional evaluations performed. I think reporting the magnitude of effect is interesting and might be interesting to have in an appendix. I also thank the authors for conducting evaluations on the Adult dataset, providing further empirical insight into the proposed fairness notion's application. Since my most significant concern regarding evaluations was addressed, I am bumping my score to Accept. However, since the authors mention "other relevant fairness notions we considered also introduced additional subtleties", it would be very helpful to discuss these initial insights somewhere. This discussion can help inform future work in this new paradigm.

---

> > > ### Author Response · Authors · 2025-08-04
> > >
> > > Thank you!
> > >
> > > We will add the additional evaluations on the magnitude of the effect and on the Adult dataset to the appendix, and will include a discussion of other fairness notions and additional future directions to the concluding section of the paper.

---

### Official Review · Reviewer_e5fy · 2025-07-01

**Clarity:** 3
**Significance:** 2
**Originality:** 2
**Rating:** 5
**Confidence:** 3

**Summary:**

If two (or more) classifiers are fair (e.g., satisfy equal opportunity), is the ecosystem containing both classifiers also fair? The authors show theoretically and empirically that fairness is not maintained in this setting.

The authors' main contributions are (1) defining what ecosystem fairness means, showing that ecosystem fairness isn’t guaranteed (2) when individual classifiers have different levels of correlation on different subpopulations or (3) when individual classifiers serve different subpopulations, and (4) empirically showing that the theoretical results hold on a real dataset.

**Questions:**

* In the experiments, was accuracy comparable across all dataset sizes? In particular, 300 samples seems like it might not be sufficient for good accuracy
* Are there different definitions of ecosystem outcomes that wouldn't reduce to ensembles with "one-yes" voting?
* The finding that large dataset sizes causes EOC to increase less is interesting. Do you have any thoughts on why this trend occurs?

**Ethical Concerns:**

["NO or VERY MINOR ethics concerns only"]

**Final Justification:**

I'm going to bump up my score to a 5- my biggest concern was the redundancy with previous work on fairness not composing, but the distinctions highlighted by the authors make sense. I'd encourage the authors to include some version of that part of the rebuttal in the camera-ready.

**Limitations:**

Yes

**Quality:**

3

**Strengths And Weaknesses:**

### Strengths
* Important problem, real-world relevance.
* Theoretical contributions are clear. The examples are helpful.
* The authors are straightforward about limitations (theoretical assumptions), but what they do present is sufficient to illustrate the general idea and additional details on generalization are in the appendix.

### Weaknesses
* It’s a well-known result that fairness does not always compose [1-2], e.g., in ensembles. The authors should cite this work and describe how their contributions are novel. The problem setups are different but the specific ecosystem framework described in the paper is very related to ensembles, just with a voting mechanism of “one-yes” rather than majority vote. Are there other instantiations of the ecosystem framework that would be distinct from ensembles? Describing these (even if it's not possible to evaluate them experimentally) would help strengthen the paper's novely.
* Clarity in Section 5 could be improved. Specifically, mention Experiment 4 in the bulleted list with the other experiments and make Figure 1 be self-contained (lines 357-359 describe it this but is hard to find)
* Experiments only looked at one dataset, and it is a dataset without a true protected attribute (“having a mortgage” feels different than race or gender, because one could make an argument that it could causally impact rather than just correlate with the outcome)


[1] Dwork et al., https://arxiv.org/abs/1806.06122

[2] Bower et al., https://arxiv.org/abs/1707.00391

---

> ### Author Rebuttal · Authors · 2025-07-29
>
> We very much appreciate your comments. Thank you!
>
> In the following, we first address your questions, and then also your comments from the "Strengths and Weaknesses" section.
>
> ### Questions
>
> On the accuracy across different dataset sizes: Accuracy improved with dataset size, and indeed, with 300 samples the accuracy was not great. What we found interesting is that our finding in this section, that the EOC-level worsens after fairness adjustment, holds for a range of accuracies, even though the effect decreases somewhat as accuracy increases.
>
> On definitions of ecosystem outcomes that don't reduce to ensembles with "one-yes" voting: Such definitions indeed exist, and, in fact, our utility-based approach provides such a definition. Although in the main text of the paper we assume that borrowers’ utility functions are one-yes – 1 if they have at least one offer, 0 otherwise – our general definition of WEC (middle of page 4) does not reduce to ensembles with one-yes voting. A particular formulation is analyzed in the appendix (Proposition 3), where we extend our results to a utility function that equals 0 if a borrower obtains no offers, 1 if she obtains 1 offer, and $k\in[0,\infty)$ if she obtains 2 offers.
>
> On the finding that large dataset sizes causes EOC to increase less: One guess about why this trend occurs is that this has to do with the classifiers’ accuracies. As the dataset size increases, the classifiers become more accurate. At the limit, when they are perfectly accurate, fairness is not really an issue – neither before nor after adjustment. Perhaps in these high-accuracy regimes the fairness adjustment does not do much, since the classifiers are already close to being fair.
>
> ### Other comments
>
> **Comment starting with "It’s a well-known result...":** Thank you for pointing out these papers. As you note, both of them also discuss the insight that fairness of individual classifiers does not imply fairness of a system composed of multiple classifiers. Our paper differs along multiple dimensions. First, in terms of motivation, these papers ask when a central platform (e.g., ad platform) will be fair when handling a task that consists of several subtasks (advertisers on the platform), where each subtask is required to be fair. Our motivation is more about competing firms in a decentralized market, where the joint activity determines the user’s utility. More specific differences are the following:
>
> [1] largely focus on individual fairness. They do have some extensions to group fairness, and their main results here are to show that there exist distributions for which individual fairness does not imply joint fairness. However, in their model the classifiers are assumed to always be independent, and they cannot capture the correlations between classifiers that lead to ecosystem unfairness. In addition, their notion of group fairness does not include EO.
>
> [2] do focus on EO, but here the main difference is that classifiers are composed sequentially: one classifier makes a prediction or decision, the outcome is then passed on to the next classifier, and so on. In our setting, in contrast, the classifiers run in parallel.
>
> Finally, although both [1] and [2] have the insight that individual fairness does not suffice for joint fairness, in our paper we also identify the forces that lead to the joint unfairness, and quantify the extent of this unfairness as a function of correlations and overlaps.
>
> **Comment starting with "Clarity in Section 5...":** Thanks, we'll make these changes.
>
> **Comment starting with "Experiments only looked at one dataset...":** Following this suggestion (also made by another reviewer), we ran the same experiments on a different dataset: The Adult Income dataset from the UC Irvine Machine Learning Repository. The dataset is smaller, and contains 48,000 instances. We ran our experiments to predict whether or not income is above \$50,000, using age, education, work hours per week, occupation, and family status as features. The protected attribute was gender. The results were broadly similar to our experiment, in terms of orders-of-magnitude. For brevity, we report here the 95%-CIs for a sample size of 3000:
>
> > Exp1:  [13.4, 20.2]
> >
> > Exp2:  [34.4, 43.0]
> >
> > Exp3:  [45.4, 54.4]
> >
> > Exp4:  [61.8, 69.8]
>
> We plan to include the full set of results in the paper.

---

> > ### Comment · Reviewer_e5fy · 2025-08-01
> >
> > thanks for addressing my comments and running the additional experiments!
> > I'm going to bump up my score to a 5- my biggest concern was the redundancy with previous work on fairness not composing, but the distinctions highlighted by the authors make sense. I'd encourage the authors to include some version of that part of the rebuttal in the camera-ready.

---

### Official Review · Reviewer_vyhi · 2025-07-02

**Clarity:** 4
**Significance:** 3
**Originality:** 2
**Rating:** 4
**Confidence:** 4

**Summary:**

This paper takes an ecosystem perspective on algorithmic fairness, focusing on how fairness is affected by competition. The paper studies this question in a stylized model for lending where there are multiple competing lenders. The key finding is that even if each lender deploys a classifier satisfying fairness, then the ecosystem may not be fair.

The paper studies in this in a stylized lending model with borrowers and two lenders. Borrowers are captured by observable features, binary protected attributes, and binary outcomes for whether they repay loans. Lenders use a classifier to determine whether to offer a loan to the borrowers based on the features and attributes. They focus on fairness for individual classifiers as given by equal opportunity and generalizations. They focus on ecosystem fairness in terms of the set of all loans available to borrowers, defining an analogue of equal opportunity.

The paper theoretically demonstrate that even if each lender’s classifier satisfies equal opportunity, the ecosystem need not satisfy equal opportunity under competition. They identify two forces: first, differences in correlations across groups, and second, differences in which borrowers are served. They then show that fairness-improving post-processing of individual classifiers can hurt equal opportunity under competition. They empirically validate these findings on the Lending Club loan dataset.

**Questions:**

- The paper does not include a discussion of how their work relates to older works that study ecosystem fairness. Can the authors discuss how their papers relate to e.g., [1] and [2]?
- Is there a different way of dividing responsibility across lenders to guarantee ecosystem fairness?

[1] Dwork and Ilvento. Fairness Under Composition. ITCS 2019.

[2] Bower, Kitche, Niss, Strauss, Vargas, and Venkatasubramanian. Fair Pipelines. FAT/ML 2017.

**Ethical Concerns:**

["NO or VERY MINOR ethics concerns only"]

**Final Justification:**

After discussions with the AC and other reviewers, I think that this paper provides an interesting analysis of how fairness is affected by competition, and I find the paper to be well-written. However, the analysis is limited to a stylized lending model, and I also didn't find the main message particularly surprising in light of prior work (even though the mechanisms/setting are different, as the authors explained in the author response). My assessment is that this paper is borderline, and I would lean slightly towards acceptance.

**Limitations:**

Yes

**Quality:**

3

**Strengths And Weaknesses:**

Strengths:
- The paper shows through a nice mix of theoretical and empirical results that there is a big disconnect between fairness for each individual classifiers and ecosystem fairness.
- The paper clearly distills two distinct forces which create this disconnect, and connects them to existing intuition. As the authors note, the first force is related to algorithmic monoculture. The second force relates to the size of each lender’s market base.
- The paper is very well written.

Weaknesses:
- The insight that fairness of individual classifiers does not guarantee ecosystem fairness has been established (in different models) in prior work. See e.g., [1] and [2]. This makes the results in this paper a bit less surprising.
- The results are limited to a stylized lending model. (However, I do not see this is a major weakness, since the key forces are clearly distilled within this model.)
- While enforcing standard fairness criteria for individual classifiers does not guarantee ecosystem fairness, it is possible that there is a different way of dividing responsibility across lenders (i.e., not using standard fairness criteria) to guarantee ecosystem fairness. The paper does not seem to discuss this possibility.

[1] Dwork and Ilvento. Fairness Under Composition. ITCS 2019.

[2] Bower, Kitche, Niss, Strauss, Vargas, and Venkatasubramanian. Fair Pipelines. FAT/ML 2017.

---

> ### Author Rebuttal · Authors · 2025-07-29
>
> Thank you for your review of the paper, and for the helpful comments!
>
> In the following, we first address your questions, and then also your comments from the "Strengths and Weaknesses" section.
>
> ### Questions
>
> * Thank you for pointing out these papers. As you note, both of them also discuss the insight that fairness of individual classifiers does not imply fairness of a system composed of multiple classifiers. Our paper differs along multiple dimensions. First, in terms of motivation, these papers ask when a central platform (e.g., ad platform) will be fair when handling a task that consists of several subtasks (advertisers on the platform), where each subtask is required to be fair. Our motivation is more about competing firms in a decentralized market, where the joint activity determines the user’s utility. More specific differences are the following:
>
>     [1] largely focus on individual fairness. They do have some extensions to group fairness, and their main results here are to show that there exist distributions for which individual fairness does not imply joint fairness. However, in their model the classifiers are assumed to always be independent, and they cannot capture the correlations between classifiers that lead to ecosystem unfairness. In addition, their notion of group fairness does not include EO.
>
>     [2] do focus on EO, but here the main difference is that classifiers are composed sequentially: one classifier makes a prediction or decision, the outcome is then passed on to the next classifier, and so on. In our setting, in contrast, the classifiers run in parallel.
>
>
>     Finally, although both [1] and [2] have the insight that individual fairness does not suffice for joint fairness, in our paper we also identify the forces that lead to the joint unfairness, and quantify the extent of this unfairness as a function of correlations and overlaps.
>
> * This is a great question. Since our motivating environment is one where lenders compete with one another, we are interested in the kinds of regulations that a regulator could implement in order to guarantee ecosystem fairness. And indeed, dividing responsibility – for example, by allowing only one lender to serve each borrower – would work. However, this would require a high level of centralization, which may not be feasible in a competitive market. In addition, we would lose out on the benefits of having multiple lenders, namely, that the possibility of multiple lenders serving the same borrower leads to fewer borrowers with $y=1$ being denied a loan.
>
>
> ### Other comments
>
> **Comment starting with "The results are limited to a stylized lending model...":** Our main modeling assumptions would also fit a setting where borrowers are job candidates, insurance buyers,  or student applicants, and lenders are employers, insurers, or colleges, respectively.

---

### Official Review · Reviewer_V2R5 · 2025-07-02

**Clarity:** 2
**Significance:** 4
**Originality:** 3
**Rating:** 5
**Confidence:** 3

**Summary:**

The paper studies the fairness of AI-based decision-making in a multi-agent system, for instance, different financial institutions offering loans or different employers hiring candidates. The paper presents theoretical and empirical results that in such systems fairness of individual systems used by different firms does not guarantee fairness of the overall ecosystem. More simply, even when the classifiers used by different banks or employers for making loan or hiring decisions satisfy fairness criteria such as equal opportunity, the overall system might not result in equal opportunity for different classes. The paper discusses two factors that may result in this behavior – the classifiers used by competing firms may have different correlations on the protected group, or the pool of candidates served by the competing firms may be overlapping, but not identical. The paper also presents experiments that show that imposing fairness adjustments to individual classifiers can worsen the fairness of the ecosystem (compared to when individual classifiers are not adjusted for fairness).

**Questions:**

Main suggestion:

1. The main paper very briefly (in the last 3 lines of the paper) alludes to the generalized results with more than two lenders and a more general utility function, which are included in the appendix. I would instead recommend that the authors bring this up earlier in the main paper. The detailed proofs can be kept in the appendix, but it would be useful to mention the results in the main paper, as it adds strength to the paper's claims. Furthermore, the empirical experiments in the paper are limited to 2 lender settings, whereas it seems useful and straightforward to try it on more than 2 lenders to see how the empirical findings generalize to a multi-agent setting with many agents.

Main questions:

2. What are the implications of the findings in the paper? The paper raises an important problem that imposing fairness in individual classifiers may degrade fairness across the ecosystem involving “competing” firms. But it is not clear what the practical takeaway should be. Should we be thinking differently about imposing a fairness adjustment? Should we refrain from making any fairness adjustments at all? Do the correlation in classifiers and overlap in applicants exhaustively cover all factors that could lead to a decline in fairness under competition, in which case, this could be used as an early signal that the fairness of the system could be compromised? I would suggest adding a discussion about the implications of the findings in the paper.

3. The result section provides an example where two somewhat biased classifiers result in EOC under competition of zero, that is, the system is fair under competition without any fairness adjustment to the individual classifiers. Was EOC=0 a coincidence, or was this experimentally designed to exhibit that two classifiers can lead to EOC=0 without adjustment, but EOC>0 after adjustment? Generally speaking, the paper discusses what conditions of unbiased classifiers result in a biased system under competition. However, can we define this relationship the other way around? Basically, generalizing the above example, what relationship between the biased classifiers would result in an unbiased system under competition?

Other questions:

4. In “fairness with a single lender” subsection where welfare-equalizing (WE) and equalized opportunity (EO) are defined, the paper mentions that EO is derived from WE by fixing $v(x, y, a, r)=1$ if $y=r=1$, and 0 otherwise, which makes sense. But then, in the definition of WE, why does r seem to be fixed to 1? Isn’t WE supposed to be general, where r can take any value between 1 to |L|?

5. The paper introduces the notation $v(x, a, y, o_1, \ldots, o_{|L|})$ as “the utility of borrower $(x, a, y)$ when he receives offers from all lenders $l$ with $o_l=1$”. What is $o_l$  here? This notation is not introduced anywhere. After staring at it for a while, I am assuming you mean that $o_l$ is whether lender $l$ makes an offer to the borrower $(x, a, y)$?

Minor formatting issue:

6. The paper mentions a Figure 1 in the results section (line 357), but this is not present in the main paper or the appendix.

**Ethical Concerns:**

["NO or VERY MINOR ethics concerns only"]

**Final Justification:**

The authors have acknowledged that my concerns regarding the need for more intuition and appropriate exposition of assumptions needed for each result can be addressed in the final version of the paper. Thus, I will maintain my positive recommendation for this paper.

**Limitations:**

Yes.

**Paper Formatting Concerns:**

No major formatting issues.

**Quality:**

3

**Strengths And Weaknesses:**

The paper discusses a very interesting and useful problem, which, to my knowledge, has not been studied previously. I will add here, though, that I am not intimately familiar with this area, but based on my search, the content of this paper seems to be novel. The theoretical and empirical results presented in the paper support their main claims well and can be impactful for thinking about fairness at the level of the overall ecosystem, rather than the fairness of individual classifiers. Overall, I believe this paper addresses an important question, provides sufficient evidence to support its claims, and provides important insights into fairness in multi-agent systems, which is applicable in many real-world scenarios.

My main concerns with the paper involve a lack of intuition and clarity of notation in a few places, which made following the theoretical results a little difficult. The examples included in the paper help understand what scenarios would correspond to different results; however, the "why" of it is not immediately clear to me. For instance, intuitively, why does a difference in correlations between classifiers result in a decline in fairness? I would recommend including more intuition along with examples and proofs, and possibly moving complete proofs to the appendix.

My other main concern with the presentation is that most assumptions and simplifications are noted in text, making it hard to track all specific assumptions that the theoretical results are based on. For instance, “For simplicity, we will assume henceforth that borrowers’ utilities depend only on the number of offers they received”. Does this mean that the results only follow in these settings, or is it only for simplifying discussion? I would strongly recommend including all assumptions in a well-defined “Assumptions” environment (similar to proposition or corollary) and then referring to specific assumptions needed for each theorem, for instance, “if Assumptions 1 and 2 hold, then …..”

---

> ### Author Rebuttal · Authors · 2025-07-29
>
> Thank you very much for all your comments and suggestions!
>
> In the following, we first address your questions, and then also your comments from the "Strengths and Weaknesses" section.
>
> ### Questions
>
> 1. These are all very helpful recommendations! We will make the generalized results more prevalent, and will add experiments involving three lenders.
> 2. Our main insight is that individual fairness is neither necessary nor sufficient for ecosystem fairness. We typically care about the latter, and so the main implication is that the ecosystem-level is the level at which we should analyze fairness. This is not to say that we should not make any fairness adjustments, but rather that the effects of adjustment on the ecosystem should be considered. Correlation and overlap are exhaustive in the sense that at least one is necessary for individual fairness to lead to unfairness in competition, and so as you note they can be used as early signals. Another way to frame it is to say that we should not complacently accept individual fairness as the end-all be-all of fairness.
> 3. Examples 3 and 4 were designed so that $\mathrm{EOC}=0$ before adjustment, in order to drive home the point that the EOC can increase after adjustment. The experimental section is meant to give a sense of the prevalence of this phenomenon. We agree that it would be an  interesting challenge to uncover the relationship between biased classifiers that lead to EOC. One thought is that it would require the group with greater correlation to also be the one that has lower false-negative rates. Finding a general characterization is beyond the scope of the current paper, but certainly a worthwhile direction for future research.
> 4. The definition of WE in that subsection looks at a setting where there is only one isolated classifier, and so $r$ is always either 1 or 0. That is, the utility of a borrower is $v(x,a,y,1)$ when getting an offer, and 0 otherwise. (We use the general utility function with $r$ for consistency with the subsequent subsection with multiple lenders.)
> 5. Yes, this is exactly the meaning. We will make it clearer.
> 6. This is a typo – it is supposed to be Table 1.
>
>
> ### Other comments
>
> **Comment starting with "My main concerns with the paper...":** Thanks for this comment. We can certainly add more intuition. For example, for the difference in correlation: under high (low) correlation, if a borrower is misclassified by one classifier, she is likely also (not) misclassified by the other. In the latter case, each deserving borrower intuitively has “two chances” to get an offer, whereas in the former case she only has “one chance”. If in one group deserving borrowers get two chances and in the other they only get one, then in the former there is a lower chance of being misclassified. This is what creates the decline in fairness.
>
> **Comment starting with "My other main concern with the presentation...":** The assumption to which you point is the one we use in the main text of the paper. The results do hold in more general settings, which we describe and analyze in the appendix. We will take your advice and be more explicit about this and other assumptions in the paper.

---

> > ### Comment · Reviewer_V2R5 · 2025-08-01
> >
> > Thank you for the clarification on my questions. As the authors mention that my main concerns will be addressed in the camera-ready version by adding more intuition and clarifying specific assumptions that each result is based on, I will commit to my positive score. Thank you for engaging with my feedback.

---

### Decision · Program_Chairs · 2025-09-17

**Decision:**

Accept (poster)

**Comment:**

## Summary

The paper seeks to model fairness "under competition". In this setting, k competing lenders decide who to offer loans to from a population. A borrower, receiving between 0 and k offers from lenders, then has to decide what to do and accrue utility based on that. The question the authors ask is:  is it possible for individual lenders to implement loan offer schemes that are fair, but that overall the outcome from borrowers in the marketplace is not. They introduce a measure of equal opportunity under competition by asking if, for each demographic group of concern, if the fraction of borrowers receiving at least one offer is more or less the same.

Their conclusions are that indeed fairness "does not aggregate": there are scenarios (and they describe two) where lenders competing in the marketplace with algorithms that look like they are fair on their own can actually lead to unfair outcomes for borrowers. They back this up with experimental evidence showing that adding fairness adjustments to individal lender algorithms could actually make things worse.

## Strengths
Reviewers generally appreciated the question being investigated, and the theoretical modeling used to answer it. The paper is relatively well written, and the theory, examples, and empirical results all reinforce each other well.

## Weaknesses
Reviewers were unconvinced that the results were surprising, and cited (almost unanimously and independently) prior work on fairness under composition that makes similar observations (namely, that when fair classifiers compose, the results will in general not be fair). The reviewers were a little concerned that the theoretical model was too stylized to be predictive of the real world, and were therefore a little concerned about the relevance of the results. One way this concern played out was in reviewer questions about how to act on the results of the paper, and what recommendations the authors would have for better design to support fairness in ecosystems.

## Author Discussion
The authors had a robust and fruitful discussion with the reviewers, where all the above points were addressed. The authors in my view were a little too hasty to dismiss the comparison to prior work. It felt like they were trying to defend their claims of novelty when the reviewers merely wanted them to place their work in the right context. I'd encourage the authors to not just insert this discussion in the conclusions (as they indicated) but actually in the related work section where it's more appropriate, and also as context setting for the broader claim of "individual fairness actions don't lead to global fairness".

In other regards, the reviewers were satisfied with the explanations provided by the authors, and the authors have incorporated changes based on the discussion.

## Justification
My overall assessment is that the paper adds a useful dimension to the broader question of how fairness methods work in an ecosystem, and their results are interesting. The reviewers shared this view, and the discussion was productive. All of these factors influence my final decision.